# Inexact Column Generation for Bayesian Network Structure Learning via Difference-of-Submodular Optimization

**Yiran Yang**
School of Data Science
The Chinese University of Hong Kong, Shenzhen
yiranyang@link.cuhk.edu.cn

**Rui Chen**
School of Data Science
The Chinese University of Hong Kong, Shenzhen
rchen@cuhk.edu.cn

## Abstract

In this paper, we consider a score-based Integer Programming (IP) approach for solving the Bayesian Network Structure Learning (BNSL) problem. State-of-the-art BNSL IP formulations suffer from the exponentially large number of variables and constraints. A standard approach in IP to address such challenges is to employ row and column generation techniques, which dynamically generate rows and columns, while the complex pricing problem remains a computational bottleneck for BNSL. For the general class of $\ell_0$-penalized likelihood scores, we show how the pricing problem can be reformulated as a difference of submodular optimization problem, and how the Difference of Convex Algorithm (DCA) can be applied as an inexact method to efficiently solve the pricing problems. Empirically, we show that, for continuous Gaussian data, our row and column generation approach yields solutions with higher quality than state-of-the-art score-based approaches, especially when the graph density increases, and achieves comparable performance against benchmark constraint-based and hybrid approaches, even when the graph size increases.

## 1 Introduction

Bayesian networks (BN) have wide applications in machine learning [27]. Its structure learning problem, i.e., identifying a directed acyclic graph (DAG) that represents the causal relationships between variables $(X_i)_{i \in V}$ from data, is a central yet challenging topic in causal inference [29]. Existing methods for learning optimal Bayesian network structure from observational data fall into three main categories [45, 40, 21]: (1) constraint-based approaches (e.g., PC [45, 14] and FCI [45, 50]) that learn the structure through a constraint satisfaction problem and involve conditional independence tests to formulate the constraints; (2) score-based approaches (e.g., hill climbing [28], dynamic programming [35, 44], integer programming [17] and machine learning approaches [49]) that evaluate candidate DAGs using scoring functions and select the DAG structure with the highest score; (3) hybrid approaches that combine both score-based and constraint-based techniques (e.g., MMHC [47]).

Score-based BNSL is known to be NP-hard, even under the assumption that each node has a restricted number of parents [13]. This inherent hardness of BNSL imposes a fundamental trade-off

between solution quality and computational efficiency. Consequently, there are two lines of work that focus on the either side: (1) exact methods that guarantee optimality but often suffer from higher computational cost and (2) heuristic approaches that offer computational efficiency but with limited optimality guarantees. Within the BNSL literature, one of state-of-the-art exact approaches to identify the highest-scoring DAG is to formulate the problem as an Integer Program (IP) [31, 15, 12], as represented by the GOBNILP method [17]. While there are different ways to formulate the IP for this problem, a standard formulation involves binary variables that indicate whether to choose the specific set as the parent set for each node, and constraints that enforce the acyclicity requirement for the DAG. Given the observational data, the objective of the IP is to maximize the score, which is a function of the potential DAG structure measuring how well the data fits the model. Solving this IP is challenging since both the number of variables and the number of constraints grow exponentially with the node size of the DAG.

Column Generation (CG) [26, 19] is a standard approach for addressing huge IPs as it effectively balances computational efficiency and solution quality. By dynamically generating variables as needed through pricing problems, CG can be used to derive efficient heuristic algorithms [32], or exact solution methods (often known as branch and price [5]) when combined with branch and cut. CG has been widely applied in operations research to solve real-world problems such as vehicle routing [20] and crew scheduling [41]. In [16, 18], Cussens introduces CG to BNSL to construct an exact branch and price algorithm. However, due to the complex pricing problems (which are formulated as mixed integer nonlinear programs in [18]), the proposed approach has limited scalability and can only be applied to solve small BNSL instances.

Indeed, to accelerate CG algorithms, it is advised that one should not try to solve all pricing problems exactly and an approximate solution often suffices until the last few iterations (for proving optimality) [37]. In this paper, built on top of [18], we advance CG solution techniques for BNSL. We develop an efficient pricing algorithm based on a reformulation of the pricing problem as a difference of submodular (DS) optimization problem, enabling efficient solution of the pricing problem via the Difference of Convex Algorithm (DCA) [22, 30]. Even though DCA does not necessarily provide global optimal solutions, it guarantees convergence and often generates high-quality candidate columns for our BNSL IP. Empirically, we demonstrate that it leads to effective primal heuristics, offering potential for developing more efficient exact solution approaches.

## 2 Preliminaries

### 2.1 Scored-based BNSL as an IP

In BNSL, a scoring function measures how well the observed data fits the DAG structure, where nodes represent individual random variables and directed edges represent direct causal dependencies. Given observational data $D$, score-based approaches aim to find a DAG with the highest score, i.e., solve the following optimization problem:

$$\max_{G} \texttt{score}(G; D) \text{ s.t. } G \text{ being acyclic,} \tag{1}$$

where $\texttt{score}(G; D)$ denotes the score of the graph $G = (V, E)$ under data $D$ (see Appendix A.1). Given a particular data type (continuous or discrete), there exist several different choices for the scoring function. In this paper, we focus on $\ell_0$-penalized likelihood scores, i.e., scoring functions of the form $\texttt{score}(G; D) = \log(L(G; D)) - \Lambda \cdot k(G)$ for some $\Lambda \geq 0$, which favor graphs with a higher maximum likelihood $L(G; D)$ and a lower number of free parameters to be estimated $k(G)$ in the graphical model. By setting $\Lambda$ differently, one can recover some commonly used scoring functions, such as the AIC score [1] and the BIC score [42].

A crucial property of $\ell_0$-penalized likelihood scores is their decomposability into node-specific components. Specifically, assuming that we have $n$ variables in our BNSL problem indexed by $i \in V = \{1, \ldots, n\}$, the score $\texttt{score}(G; D)$ of a DAG $G$ defined over nodes $V$ satisfies

$$\texttt{score}(G; D) = \sum_{i=1}^{n} \texttt{score}_i \left( \texttt{pa}_i(G) \right),$$

where $\texttt{score}_i(\texttt{pa}_i(G))$ denotes the local score of node $i$ (where we omit its dependence on $D$ for simplicity), whose value only depends on the parent set $\texttt{pa}_i(G)$ of node $i$ in graph $G$, i.e., the set

of nodes pointing towards node $i$ in the directed graph $G$. Specific definitions of the $\ell_0$-penalized likelihood score and its local version, in both discrete and continuous data cases, can be found in Appendix A.1.

The additive nature of the scoring function enables us to reformulate (1) as an integer (linear) program. A classic IP formulation based on cluster constraints [15, 31] is as follows

$$\max_x \quad \sum_{i=1}^n \sum_{J \in \mathcal{P}_i} \mathtt{score}_i(J) \cdot x_{i \leftarrow J} \tag{2a}$$

$$\text{s.t.} \quad \sum_{J \in \mathcal{P}_i} x_{i \leftarrow J} = 1, \qquad\qquad i = 1, \ldots, n, \tag{2b}$$

$$\sum_{i \in C} \sum_{\substack{J \in \mathcal{P}_i: \\ J \cap C \neq \emptyset}} x_{i \leftarrow J} \leq |C| - 1, \quad C \in \mathcal{C}, \tag{2c}$$

$$x_{i \leftarrow J} \in \{0, 1\}, \qquad\qquad i = 1, \ldots, n, \ J \in \mathcal{P}_i. \tag{2d}$$

Here $\mathcal{P}_i := 2^{V \setminus \{i\}}$ denotes the set of candidate parent sets of node $i$, the binary decision variable $x_{i \leftarrow J}$ indicates whether parent set $J$ is selected for node $i$, with an associated local score $\mathtt{score}_i(J)$. The linear objective reflects the score of the graph composed of the selected parent sets. Constraint (2b) enforces exactly one parent set to be chosen per node, while cluster constraints (2c) [31] guarantee acyclicity by preventing directed cycles among any nonempty node subset $C \in \mathcal{C} := 2^V \setminus \{\emptyset\}$.

Note that formulation (2) involves $\Theta(n2^{n-1})$ binary variables and $\Theta(2^n)$ constraints. Due to its exponential size, even directly writing down (2) as an IP in a computer is infeasible for reasonably large $n$, not to say solving it. Although the exponential number of constraints can be addressed through row generation (i.e., replacing $\mathcal{C}$ by a small subset $\hat{\mathcal{C}}$ in (2) and iteratively adding violated ones into $\hat{\mathcal{C}}$, see Appendix B) [15], the exponential number of variables remains a significant challenge. To deal with exponentially many variables, existing BNSL IP approaches often rely on a (compromising) assumption that the size of the parent set for each node is no larger than some constant $k$, in which case one may replace $\mathcal{P}_i$ by $\{J \in \mathcal{P}_i : |J| \leq k\}$ in (2) for all $i$ to reduce the number of variables [6]. Instead of putting such a restriction on the parent sets, we consider CG for BNSL as proposed in [16, 18].

## 2.2 Column Generation and the Pricing Problem for BNSL

Rather than including all the variables, CG handles IPs with huge number of variables by dynamically generating only the necessary ones through the so-called pricing problems. In the context of the BNSL IP (2), as demonstrated in [18], the CG procedure begins with a restricted set of parent sets $\hat{\mathcal{P}}_i \subseteq \mathcal{P}_i$ for each node $i = 1, \ldots, n$, and considers a restricted version of (2) only over variables $x_{i \leftarrow J}$, for $i = 1, \ldots, n, \ J \in \hat{\mathcal{P}}_i$, i.e., (2) with a small subset of columns. We refer to the linear programming (LP) relaxation of this restricted problem as the *Restricted Master LP* (RMLP). Based on the current RMLP solution, CG iteratively searches for additional parent set choices that may improve the objective value through the solution of a pricing problem for each node $i$, and add them to $\hat{\mathcal{P}}_i$. This pricing problem aims to optimize the so-called reduced costs, to identify high-quality parent set choices to gradually enlarge the variable space and solve the LP relaxation of (2).

As pointed out in [18], the pricing problem for generating candidate parent sets for node $i$ can be formulated as the following optimization problem:

$$\min_{J \in \mathcal{P}_i} z_i(J; \lambda^*) := -\mathtt{score}_i(J) + \lambda_i^* + \sum_{\substack{C \in \hat{\mathcal{C}}: \\ i \in C, \ J \cap C \neq \emptyset}} \lambda_C^*, \tag{3}$$

where $\lambda_i^*$ and $\lambda_C^* \geq 0$ are the optimal dual values associated with constraints (2b) and (2c) in the RMLP, respectively. The value of $z_i(J; \lambda^*)$ is known as the reduced cost [8] of the variable $x_{i \leftarrow J}$. Its negative measures how much the RMLP objective value might increase by adding the column associated with $x_{i \leftarrow J}$ to RMLP. The column generation process terminates when the optimal reduced cost (3) becomes nonnegative for all the nodes. In this paper, instead of using CG to construct an exact solution approach like branch and price, we consider a straightforward CG-based heuristic named the restricted master heuristic [32], where we solve the IP (2) with the candidate set $\hat{\mathcal{P}}_i$ generated by CG, namely the *Restricted Master IP* (RMIP), after CG (approximately) terminates.

---

**Algorithm 1** The Difference of Convex Algorithm

---

1: **Input**: Convex functions $g^L$, $f^L$ (and $z^L = g^L - f^L$), initial point $x^0$, threshold $\epsilon$
2: Initialize $t \leftarrow 0$
3: **repeat**
4:     $y^t \in \partial f^L(x^t)$
5:     $x^{t+1} \leftarrow \arg\min_x \{g^L(x) - \langle y^t, x \rangle\}$
6:     $t \leftarrow t + 1$
7: **until** $|z^L(x^t) - z^L(x^{t-1})| < \epsilon$
8: **Return** Solution $x^t$, objective value $z^L(x^t)$

---

## 2.3 Difference of Submodular Optimization via DCA

The pricing problem (3) is a set function optimization problem. Although problem (3), in the continuous Gaussian case, can be formulated as a mixed-integer nonlinear program (MINLP), solving it with an MINLP solver is highly computationally demanding as shown in [18]. Taking a direct set function optimization perspective, it is known that any set function can be written as a difference of submodular (DS) functions although finding such a DS decomposition has exponential complexity [30]. However, when such a decomposition is known, one can develop efficient algorithms to find local solutions [30, 22]. Here we briefly review a DS optimization algorithm [22] based on the well-known DCA [2].

A DS function $z : 2^V \to \mathbb{R}$ is a set function that can be expressed as:

$$z(J) = g(J) - f(J),$$

where both $g : 2^V \to \mathbb{R}$ and $f$ are submodular set functions (see Appendix A.2) [25]. For a set function $h : 2^V \to \mathbb{R}$, its minimization can be equivalently formulated through its Lovász extension $h^L : [0,1]^n \to \mathbb{R}$ (see Appendix A.2) [36]. The Lovász extensions for submodular functions are convex [36], enabling the reformulation of $\min_J z(J) = g(J) - f(J)$ as a difference of convex (DC) program

$$\min_x z^L(x) = g^L(x) - f^L(x).$$

To solve this DC optimization problem, DCA (Algorithm 1) iterates between two key steps. First, it computes a linear approximation $\widetilde{f}^L(x)$ of $f^L(x)$ at the current iteration point $x^t$ defined by

$$\widetilde{f}^L(x) = f^L(x^t) + \langle y^t, x - x^t \rangle,$$

where $y^t$ is a subgradient of $f^L(x)$ at $x^t$. Subsequently, it minimizes a convex approximation of the objective $z^L(x)$, with $f^L(x)$ replaced by $\widetilde{f}^L(x)$. This subproblem is solved to obtain the next iteration point $x^{t+1}$.

As the subgradient of the Lovász extension of a submodular function can be efficiently evaluated through its function evaluation, the convex program (Line 5) in Algorithm 1 can be efficiently solved via bundle methods [4]. In our case, we adopt the classic Kelley's Algorithm [33] for numerical experiments.

One important advantage of DCA is that it guarantees a non-increasing sequence of objective values, i.e., for $t \geq 0$, we have

$$z^L(x^{t+1}) \leq z^L(x^t). \tag{4}$$

It is also easy to recover a solution of the original set function optimization problem from DCA, as the function value of $z$ agrees with $z^L$ at integer solutions. Since the convex function $g^L(x) - \langle y^t, x \rangle$ is the Lovász extension (which characterizes the convex envelope) of a submodular function (i.e., $g$ plus a modular function), all extreme point solutions of the convex program (Line 5) in Algorithm 1 are integer solutions.

## 3 Solution of the Pricing Problem

### 3.1 The Pricing Problem as DS Optimization

A key step in CG is the solution of the pricing problem (3) that iteratively selects "promising" columns to add into the restricted problem formulation. Existing MINLP approaches are known

to have very limited scalability to solve the pricing problem for BNSL [18]. We show in this section how the pricing problem can be explicitly rewritten as a DS optimization problem, enabling us to take advantage of the DCA algorithm we describe in Section 2.3.

Recall that the pricing problem for node $i$ is formulated as minimizing the reduced cost

$$\min_{J \in \mathcal{P}_i} z_i(J; \lambda^*) = -\texttt{score}_i(J) + \lambda_i^* + \sum_{\substack{C \in \hat{\mathcal{C}}: \\ i \in C, \, J \cap C \neq \emptyset}} \lambda_C^*.$$

For both continuous and discrete cases, we show how the $\ell_0$-penalized likelihood score $z_i(J; \lambda^*)$ can be expressed as a DS function.

**Proposition 1** (Continuous Case). *For continuous data with the $\ell_0$-penalized Gaussian likelihood score, $z_i(J; \lambda^*)$ satisfies*

$$z_i(J; \lambda^*) = g(J) - f(J) + \frac{N}{2} \log(2\pi + 1),$$

*where*

$$g(J) = \frac{N}{2} \log \det(\hat{\Sigma}_{J \cup i, J \cup i}) + \Lambda |J| + \sum_{\substack{C \in \hat{\mathcal{C}}: \\ i \in C, \, J \cap C \neq \emptyset}} \lambda_C^* + \lambda_i^*$$

*and*

$$f(J) = \frac{N}{2} \log \det(\hat{\Sigma}_{J,J})$$

*are both submodular functions, with $\hat{\Sigma}_{J,J}$ and $\hat{\Sigma}_{J \cup i, J \cup i}$ denoting the empirical covariance matrices associated with variables $\{X_j : j \in J\}$ and $\{X_j : j \in J \cup \{i\}\}$, respectively.*

**Proposition 2** (Discrete Case). *For discrete data with the $\ell_0$-penalized multinomial likelihood score, $z_i(J; \lambda^*)$ satisfies*

$$z_i(J; \lambda^*) = g(J) - f(J),$$

*where*

$$g(J) = N \cdot H(J \cup \{i\}) + \sum_{\substack{C \in \hat{\mathcal{C}}: \\ i \in C, \, J \cap C \neq \emptyset}} \lambda_C^* + \lambda_i^*$$

*and*

$$f(J) = N \cdot H(J) - \Lambda(a_i - 1) \prod_{j \in J} a_j$$

*are both submodular functions, with $H(J)$ and $H(J \cup i)$ denoting the joint entropy associated with variables $\{X_j : j \in J\}$ and $\{X_j : j \in J \cup \{i\}\}$, respectively, and $a_j$ denoting the arity (i.e., the number of possible values it can take) of the variable $X_j$ for $j = 1, \dots n$.*

Proofs of Propositions 1 and 2 can be found in Appendix C.

### 3.2 Implementation of DCA for Pricing

We have reformulated the pricing objective as a DS function in Propositions 1 and 2. As established in Section 2.3, minimizing the pricing objective is then equivalent to minimizing the difference of convex functions $z^L(x) = g^L(x) - f^L(x)$, where $g^L(x)$ and $f^L(x)$ represent the Lovász extension of submodular functions $g$ and $f$, respectively.

To optimize this objective, we apply DCA, i.e., Algorithm 1, to iteratively minimize $z^L(x)$ through its successive convex approximations. The DCA procedure for solving the pricing problem for node $i$ proceeds as follows. The algorithm begins by initializing an $(n-1)$-dimensional solution vector $(x_j)_{j \in V \setminus \{i\}}$. The required inputs include the dataset $D \in \mathbb{R}^{N \times n}$, node index $i$, optimal dual solutions $\lambda^*$ of RMLP, regularization parameter $\Lambda$, and convergence thresholds $\epsilon$.

During iteration $t$, the algorithm first computes the subgradient $y^t$ of $f^L$ at the current solution $x^t$. Without loss of generality, assume for simplicity that $i = n$. Let $\sigma$ be a permutation of $\{1, \dots, n-1\}$

such that $x^t_{\sigma(1)} \geq x^t_{\sigma(2)} \geq \ldots \geq x^t_{\sigma(n-1)}$. For the continuous case, a subgradient $y^t$ of $f^L$ at $x^t$ is given by

$$
y^t_{\sigma(k)} = \begin{cases} \log \det(\hat{\Sigma}_{\{\sigma(1)\}}) & k = 1 \\ \log \det(\hat{\Sigma}_{\{\sigma(1),\ldots,\sigma(k)\}}) - \log \det(\hat{\Sigma}_{\{\sigma(1),\ldots,\sigma(k-1)\}}) & k \geq 2, \end{cases}
$$

where $\hat{\Sigma}_{\{\sigma(1),\ldots,\sigma(k)\}}$ denotes the empirical covariance matrix for the corresponding columns $\{\sigma(1),\ldots,\sigma(k)\}$ of the data matrix $D \in \mathbb{R}^{N \times n}$.

To improve computational efficiency, we compute subgradients using Cholesky decomposition. By the lemma below, for any fixed permutation $\sigma$, we can efficiently calculate $\log \det(\hat{\Sigma}_{\sigma(1),\ldots,\sigma(m)})$ ($m < n - 1$) directly from the Cholesky factor $L$ of the permuted matrix $\hat{\Sigma}_{\sigma(1),\ldots,\sigma(n-1)} = LL^\top$. This approach requires only a single Cholesky decomposition of the permuted covariance matrix to obtain all necessary subgradients for the Lovász extension, rather than explicitly computing $\log \det(\hat{\Sigma}_{(\cdot)})$ for $O(n)$ times.

**Lemma 1.** *Let $M$ be a $p \times p$ positive definite symmetric matrix with lower-triangular Cholesky factor $L$. In block form, these matrices can be written as*

$$
M = \begin{bmatrix} M_{11} & M_{12} \\ M_{21} & M_{22} \end{bmatrix}, \quad L = \begin{bmatrix} L_1 & O \\ Y & L_2 \end{bmatrix},
$$

*where $M_{11}$ and $L_1$ are $m \times m$ blocks ($m < p$) and $O$ is a zero matrix of appropriate size. Then $L_1$ is the Cholesky factor of $M_{11}$, and consequently,*

$$
\log \det(M_{11}) = \log(\det(L_1)^2) = 2 \sum_{k=1}^{m} \log L_{k,k}.
$$

For the discrete case, a subgradient $y^t$ of $f^L$ at $x^t$ is given by

$$
y^t_{\sigma(k)} = \begin{cases} H(\{\sigma(1)\}) - \Lambda(a_n - 1)(a_{\sigma(1)} - 1) & k = 1 \\ H(\{\sigma(1),\ldots\sigma(k)\}) - H(\{\sigma(1),\ldots,\sigma(k-1)\}) - \Lambda(a_n - 1)(a_{\sigma(k)} - 1)\prod_{j=1}^{k-1} a_{\sigma(j)} & k \geq 2. \end{cases}
$$

After evaluating $\log \det(\hat{\Sigma}_{(\cdot)})$ or $H(\cdot)$ during the computation of $y^t$, it is crucial to save the evaluated values to avoid repeating calculations in the future. This small trick greatly improves the efficiency of DCA in pricing problem solutions in practice.

## 4 Row and Column Generation Scheme

Algorithm 2 demonstrates how we incorporate our DCA pricing method into a simultaneous row and column generation framework. The algorithm begins by initializing the candidate parent sets $\hat{\mathcal{P}}_{1:n}$ and the cluster set $\hat{\mathcal{C}}$ with basic candidates (Line 2). It then iterates among three main phases: (1) a CG phase (Lines 4-14) where the RMLP is solved to obtain optimal dual solutions $\lambda^*$, followed by the solution of the pricing problem for each node via DCA — The pricing solutions that have negative reduced costs are added to $\hat{\mathcal{P}}_i$. The CG phase repeats until all reduced costs become non-negative. In Line 8, the pricing problem on node $i$ is solved only when the current reduced cost $rc_i$ is negative, to avoid spending too much time on pricing; (2) a row generation phase (Lines 15-21) where the RMLP optimal solution is used to identify the most violated cluster constraint (see Appendix B for details) which is added to cluster set $\hat{\mathcal{C}}$; (3) the integer solution phase (Line 22), where the algorithm finds the optimal DAG $G$ with current candidate $\hat{\mathcal{P}}_{1:n}$ — This is achieved by solving RMIP using branch and cut with callbacks (a feature that allows users to interrupt the solution process in modern IP solvers such as Gurobi), where we check if any cluster constraint is violated when we meet an integer incumbent and add violated cluster constraints as lazy constraints. The three phases iterate until convergence to a valid DAG solution, progressively refining both the columns in $\hat{\mathcal{P}}_{1:n}$ and cluster constraints in $\hat{\mathcal{C}}$.

When we implement the CG phase in Algorithm 2, in addition to the final solution generated by DCA, we add to $\hat{\mathcal{P}}_i$ all candidate parent sets associated with intermediate $x^t$ solutions that have negative reduced costs. Empirically, we find that it helps accelerate convergence and improves solution quality. In addition, we initialize the RMIP with the optimal IP solution obtained from the last round, as it remains feasible and empirically reduces the solution time.

---

**Algorithm 2** Row and Column Generation for BNSL

---

1: **Input**: Data matrix $D \in \mathbb{R}^{N \times n}$, regularization parameter $\Lambda$
2: Initialize the cluster set $\hat{\mathcal{C}} \leftarrow \emptyset$, and candidate parent sets $\hat{\mathcal{P}}_i \leftarrow \{\emptyset\}$ for $i = 1, \ldots, n$
3: **repeat**
4:     Initialize $\texttt{rc}_i \leftarrow -\infty$ for $i = 1, \ldots, n$              ▷ initialize reduced costs
5:     **repeat**
6:         Solve RMLP to obtain the optimal dual values $\lambda^*$
7:         **for** $i = 1, \ldots, n$ **do**
8:             **if** $\texttt{rc}_i < 0$ **then**              ▷ selectively solve pricing problems
9:                 Choose an initial point $x^0$ for DCA
10:                 $(\texttt{pa}, \texttt{rc}_i) \leftarrow \text{DCA-Pricing}(D, i, x^0, \lambda^*, \Lambda, \epsilon)$       ▷ DCA pricing
11:                 $\hat{\mathcal{P}}_i \leftarrow \hat{\mathcal{P}}_i \cup \{\texttt{pa}\}$             ▷ column generation
12:             **end if**
13:         **end for**
14:     **until** $\texttt{rc}_i \geq 0$ for $i = 1 \ldots n$
15:     **repeat**
16:         Solve RMLP with updated $\hat{\mathcal{P}}$ and $\hat{\mathcal{C}}$
17:         Identify the most violated cluster constraint $C^*$        ▷ row generation for RMLP
18:         **if** the cluster constraint associated with $C^*$ is violated **then**
19:             $\hat{\mathcal{C}} \leftarrow \hat{\mathcal{C}} \cup \{C^*\}$
20:         **end if**
21:     **until** no new violated constraint can be found
22:     $G \leftarrow$ optimal DAG chosen from $\hat{\mathcal{P}}_{1:n}$; simultaneously update $\hat{\mathcal{C}}$ ▷ row generation for RMIP
23: **until** $G$ converges
24: **return** $G$

---

## 4.1 Choice of the Initial Point

A step that remains ambiguous in Algorithm 2 is the choice of the initial point $x^0$ for DCA (Line 9), which we find having a significant impact on the overall solution time and solution quality. Below we first describe two initialization strategies.

The first strategy is a warm-start initialization strategy. Note that by the complementarity property of optimal solutions in LP [8], the parent sets selected (i.e., parent sets $J \in \hat{\mathcal{P}}_i$ associated with the primal variables $x_{i \leftarrow J}$ taking positive values) in the RMLP solution have reduced costs equal to zero. Therefore, we consider using the parent sets selected in the RMLP solution to construct the initial solution $x^0$ for DCA. Due to the monotonicity property (4) of DCA, it often quickly finds columns with negative reduced costs, leading to a rapid convergence of DCA. However, a drawback of the warm-start initialization strategy is that it may restrict exploration of parent sets as the evaluation remains localized around columns already contained in the candidate parent set choice set. Preliminary experiments show that a pure warm-start initialization strategy can quickly find a DAG with a good score while the solution often deviates a lot from the ground truth due to its greedy nature.

The second strategy employs a random initialization that samples a point $x^0$ uniformly from $[0, 1]^{V \setminus \{i\}}$. This random initialization strategy facilitates global exploration on the parent set space and may discover patterns distant from the current ones in $\hat{\mathcal{P}}_i$. However, we find that exclusive reliance on random initialization can significantly slow down DCA convergence, as computational resources may be expended on evaluating solutions far from the optimum.

A hybrid initialization strategy that combines efficiency and diversity of the previous two strategies is often more effective in practice. In our implementation, we initially employ random initialization to broadly explore the solution space. Once the candidate set $\hat{\mathcal{P}}_i$ contains a reasonably large number (set to be 50 in our implementation) of parent sets, we switch to warm-start initialization for pricing at node $i$. This hybrid approach focuses on local refinement around the current best pattern while building upon the foundation of global exploration. A comparison of the three initialization strategies can be found in Appendix D.1.

Table 1: Score and Time Comparison between CG-DCA and GOBNILP on Gaussian Datasets

| $(n, N, d)$ | BIC Score Gap (%) | | Time (seconds) | |
|---|---|---|---|---|
| | CG-DCA | GOBNILP | CG-DCA | GOBNILP |
| (15,  5000, 0.5) | (0.00, 0.00) **[0.00]** | (0.00,  1.46) [0.15] | (  1.12,     14.71) [   **4.06**] | (   5.47,    24.28) [   8.33] |
| (15,  5000, 1.0) | (0.00, 0.24) **[0.03]** | (0.00,  1.06) [0.22] | (  2.54,     93.58) [  25.76] | (   5.72,    56.11) [  **19.74**] |
| (15,  5000, 1.5) | (0.00, 0.62) **[0.16]** | (0.00,  2.18) [0.62] | ( 17.95,    196.01) [  76.72] | (  11.51,   137.52) [  **60.09**] |
| (15,  5000, 2.0) | (0.00, 0.84) **[0.39]** | (0.48,  4.88) [2.06] | ( 41.13,    340.22) [ 170.39] | (  22.99,   641.90) [ **167.41**] |
| (15, 20000, 0.5) | (0.00, 0.00) **[0.00]** | (0.00,  0.64) [0.06] | (  3.53,     35.11) [  10.17] | (   5.53,    12.62) [   **7.35**] |
| (15, 20000, 1.0) | (0.00, 0.27) **[0.03]** | (0.00,  1.19) [0.23] | (  7.52,    100.20) [  36.71] | (   1.31,    93.02) [  **19.67**] |
| (15, 20000, 1.5) | (0.00, 0.11) **[0.02]** | (0.00,  3.10) [0.86] | ( 25.17,    265.33) [ 104.25] | (  12.34,   285.39) [  **76.29**] |
| (15, 20000, 2.0) | (0.00, 0.94) **[0.23]** | (0.98,  3.97) [2.01] | ( 55.11,    335.96) [ 229.80] | (  11.66,   190.19) [  **89.92**] |
| (20,  5000, 0.5) | (0.00, 0.01) **[0.00]** | (0.00,  0.18) [0.02] | (  2.49,     20.20) [   9.53] | (   1.36,     6.99) [   **3.38**] |
| (20,  5000, 1.0) | (0.00, 0.08) **[0.01]** | (0.00,  1.38) [0.35] | (  5.97,    150.18) [  **84.34**] | (   1.59,  1714.57) [ 287.87] |
| (20,  5000, 1.5) | (0.01, 0.51) **[0.22]** | (0.00,  4.30) [1.63] | (144.94,    539.72) [ **312.67**] | ( 121.01,  3483.96) [ 962.50] |
| (20,  5000, 2.0) | (0.22, 1.15) **[0.70]** | (0.07, 10.42) [4.77] | (403.98,   1904.24) [**1039.53**] | (1063.89,  8703.57) [4603.81] |
| (20, 20000, 0.5) | (0.00, 0.00) **[0.00]** | (0.00,  0.35) [0.04] | (  6.57,     36.37) [  18.42] | (   1.50,    11.25) [   **4.92**] |
| (20, 20000, 1.0) | (0.00, 0.22) **[0.03]** | (0.00,  1.49) [0.22] | ( 13.72,    330.05) [ **133.69**] | (   1.54,  1371.66) [ 261.89] |
| (20, 20000, 1.5) | (0.00, 0.43) **[0.15]** | (0.00,  3.17) [1.45] | (216.26,    821.01) [ **425.17**] | ( 135.59,  7269.60)[2457.26] |
| (20, 20000, 2.0) | (0.09, 1.08) **[0.54]** | (0.21,  7.05) [3.63] | (457.33,   1739.24) [ **997.99**] | ( 517.67, 10801.59) [4394.45] |
| (25,  5000, 0.5) | (0.00, 0.00) **[0.00]** | (0.00,  0.82) [0.12] | (  2.73,     67.05) [  21.97] | (   1.61,   151.24) [  **21.95**] |
| (25,  5000, 1.0) | (0.00, 0.25) **[0.03]** | (0.00,  1.83) [0.56] | ( 28.04,    953.40) [ 261.84] | (  11.14,  7451.37) [1964.11] |
| (25,  5000, 1.5) | (0.00, 1.11) **[0.24]** | (0.09,  5.27) [1.92] | (144.19,   2880.93) [ **949.90**] | (  84.46, 10801.61) [5805.01] |
| (25,  5000, 2.0) | (0.00, 2.18) **[0.86]** | (1.84, 10.18) [5.15] | (448.93, 12665.38) [**3054.63**] | ( 723.38, 10801.63) [8500.77] |
| (25, 20000, 0.5) | (0.00, 0.01) **[0.00]** | (0.00,  0.78) [0.09] | (  9.50,    164.57) [  50.22] | (   1.62,   169.94) [  **23.96**] |
| (25, 20000, 1.0) | (0.00, 0.22) **[0.06]** | (0.00,  2.72) [0.40] | ( 70.34,   1287.17) [ **404.43**] | (  11.86, 10801.48) [1948.05] |
| (25, 20000, 1.5) | (0.00, 1.19) **[0.23]** | (0.66,  5.47) [2.21] | (237.86,   4955.58) [**1372.59**] | ( 115.63, 10801.73) [5853.91] |
| (25, 20000, 2.0) | (0.00, 2.71) **[0.67]** | (1.50,  8.30) [4.51] | (733.04,   7371.52) [**3189.98**] | (2302.81, 10801.71) [7967.28] |

*The values are the (min, max)[average] over ten independent instances.

# 5  Numerical Experiments

To evaluate the efficiency of our method, we conducted numerical experiments comparing the performance of our Column Generation with DCA-pricing (CG-DCA) method against five baseline methods: GOBNILP [17], CG-MINLP [18] (in Section 5.1), HC [28] (in Appendix D.2), stable-PC [14] and MMHC [47]. For score-based approaches, we use the BIC score as our scoring function. While CG-MINLP is evaluated on `Gaussian.test` dataset in the R package `bnlearn` [43], other methods are tested on both simulated and real-world datasets. We implement CG-DCA in python, and utilize the python implementation of GOBNILP. The last three baseline methods are tested using `bnlearn` in R. All IP-related experiments are conducted on a Linux machine with two Intel XEON Platinum 8575C processors, with the number of threads used limited to 1. All IPs are solved using Gurobi 11.0.3.

## 5.1  Comparing DCA with MINLP for Pricing

We first compare the performance of CG with different pricing algorithms: our proposed CG-DCA versus the exact MINLP solver for pricing (CG-MINLP). The evaluation is conducted on the `Gaussian.test` dataset containing a ground truth graph with 7 nodes and 7 edges. For this small instance, CG-DCA successfully learns the exact ground truth structure in 1.2 seconds, while CG-MINLP requires 19.2 seconds to achieve the same result.

For larger instances, CG-MINLP usually fails to produce a valid solution DAG within a reasonable time limit as solving the pricing problem exactly as an MINLP is quite cumbersome in practice. For a graph with node size $n = 20$ and average in-degree $d = 1$ ($N = 5,000$, simulated as in Section 5.2), solving the first pricing problem using MINLP takes an average of 35.1 seconds per node, while the DCA method takes 0.3 second per node for the same instance. The difference is even larger for later pricing problems. In addition, MINLP approaches suffer from numerical issues as they require an approximation for the nonlinear $\log$ function (at least in Gurobi), producing inaccurrate estimates for the pricing objective.

## 5.2  Results on Simulated Gaussian Data

Now we test on larger instances with Gaussian data. Following the experimental setup in [18, 9], we randomly generate Bayesian networks with node size $n \in \{15, 20, 25\}$, and simulate Gaussian data from those networks. To generate the DAGs, we first fix a topological order of the nodes

Table 2: Performance Comparison of CG-DCA, GOBNILP (GI), (Stable-)PC and MMHC on Gaussian Datasets

| $(n, N, d)$ | Precision | | | | Recall | | | | SHD | | | |
|---|---|---|---|---|---|---|---|---|---|---|---|---|
| | CG-DCA | GI | PC | MMHC | CG-DCA | GI | PC | MMHC | CG-DCA | GI | PC | MMHC |
| (15,  5000, 0.5) | **0.78** | 0.66 | 0.59 | 0.50 | **0.80** | 0.73 | 0.62 | 0.51 | **1.70** | 2.90 | 3.00 | 3.50 |
| (15,  5000, 1.0) | 0.70 | 0.67 | **0.72** | 0.59 | **0.78** | 0.72 | 0.67 | 0.54 | 6.20 | 5.00 | **4.80** | 6.70 |
| (15,  5000, 1.5) | 0.65 | **0.73** | 0.61 | 0.56 | 0.77 | **0.80** | 0.54 | 0.48 | 11.80 | **7.40** | 11.50 | 12.20 |
| (15,  5000, 2.0) | 0.55 | **0.58** | 0.54 | 0.52 | **0.72** | 0.65 | 0.39 | 0.36 | 21.90 | **16.90** | 20.90 | 20.40 |
| (15, 20000, 0.5) | **0.71** | 0.69 | 0.68 | 0.53 | **0.74** | 0.73 | 0.68 | 0.53 | 2.20 | 2.20 | **1.90** | 3.00 |
| (15, 20000, 1.0) | 0.76 | **0.78** | 0.71 | 0.61 | 0.78 | **0.81** | 0.69 | 0.59 | 4.10 | **4.00** | 4.70 | 6.20 |
| (15, 20000, 1.5) | **0.76** | 0.73 | 0.72 | 0.66 | **0.85** | 0.79 | 0.66 | 0.57 | **6.90** | 9.40 | 8.60 | 10.00 |
| (15, 20000, 2.0) | **0.61** | 0.53 | 0.53 | 0.55 | **0.75** | 0.60 | 0.43 | 0.39 | 20.60 | 19.90 | 20.01 | **19.80** |
| (20,  5000, 0.5) | 0.74 | 0.71 | **0.76** | 0.59 | **0.80** | 0.75 | 0.79 | 0.60 | 3.40 | 3.50 | **2.60** | 4.40 |
| (20,  5000, 1.0) | 0.71 | 0.69 | **0.75** | 0.65 | 0.76 | **0.77** | 0.71 | 0.59 | **3.70** | 7.50 | 5.90 | 7.90 |
| (20,  5000, 1.5) | 0.50 | 0.61 | 0.62 | **0.65** | **0.71** | **0.71** | 0.50 | 0.49 | 28.20 | **16.80** | 17.90 | 16.80 |
| (20,  5000, 2.0) | 0.44 | 0.46 | 0.54 | **0.55** | **0.66** | 0.53 | 0.34 | 0.31 | 36.40 | 33.60 | 31.50 | **31.10** |
| (20, 20000, 0.5) | **0.82** | 0.80 | 0.73 | 0.58 | **0.83** | 0.82 | 0.76 | 0.59 | **2.20** | 2.60 | 3.00 | 4.60 |
| (20, 20000, 1.0) | 0.61 | 0.73 | **0.79** | 0.68 | 0.70 | **0.79** | 0.77 | 0.65 | 8.30 | 6.90 | **4.90** | 7.10 |
| (20, 20000, 1.5) | **0.65** | 0.58 | 0.64 | 0.63 | **0.81** | 0.68 | 0.56 | 0.50 | 16.70 | 18.20 | **16.00** | 16.50 |
| (20, 20000, 2.0) | 0.52 | 0.48 | 0.51 | **0.56** | **0.73** | 0.55 | 0.37 | 0.33 | 33.00 | 32.40 | 31.30 | **29.30** |
| (25,  5000, 0.5) | 0.82 | 0.61 | **0.84** | 0.69 | **0.84** | 0.67 | **0.84** | 0.69 | **2.70** | 5.90 | **2.70** | 4.40 |
| (25,  5000, 1.0) | **0.81** | 0.69 | 0.80 | 0.74 | **0.86** | 0.76 | 0.75 | 0.69 | **6.70** | 11.10 | 7.40 | 8.70 |
| (25,  5000, 1.5) | 0.66 | 0.57 | 0.70 | **0.71** | **0.75** | 0.68 | 0.58 | 0.56 | **13.60** | 22.10 | 18.50 | 18.00 |
| (25,  5000, 2.0) | 0.54 | 0.47 | 0.53 | **0.59** | **0.73** | 0.54 | 0.35 | 0.35 | 45.70 | 40.30 | 38.70 | **36.50** |
| (25, 20000, 0.5) | 0.76 | 0.80 | **0.81** | 0.70 | 0.80 | **0.84** | 0.83 | 0.70 | 3.70 | **2.80** | **2.80** | 4.30 |
| (25, 20000, 1.0) | 0.76 | 0.71 | **0.81** | 0.68 | **0.86** | 0.76 | 0.78 | 0.64 | 9.10 | 9.90 | **6.70** | 9.90 |
| (25, 20000, 1.5) | 0.57 | 0.53 | **0.73** | 0.63 | **0.67** | 0.66 | 0.63 | 0.51 | **14.10** | 25.00 | 16.50 | 19.50 |
| (25, 20000, 2.0) | 0.54 | 0.48 | **0.61** | 0.59 | **0.66** | 0.56 | 0.46 | 0.40 | **23.90** | 39.10 | 33.80 | 34.10 |

*Each value is the average over ten independent instances.

and then simulate edges according to the specified probabilities where edges must be directed from lower-order nodes to higher-order nodes (there are $n(n-1)/2$ such possible edges). While we do not enforce a maximum in-degree of each node, we randomly simulate edges using their existence probability $p$ determining the overall density of the graph, i.e., for each possible edge we include it in the network with probability $p$. The average in-degree $d$ of nodes in the ground truth DAG is controlled by the edge existence probability since $p = 2d/(n-1)$. We vary $p$ such that $d$ is varied among $\{0.5, 1, 1.5, 2\}$.

After generating the random graph structure, we simulate Gaussian data based on the network topology. The linear coefficients for the features were randomly drawn from the interval $\pm[0.5, 2]$, with additive noise following the Normal distribution $N(0, \sigma^2)$. The true value of variance parameter $\sigma^2$ is uniformly drawn from the interval $[0.7, 1.2]$.

Tables 1 and 2 present a comparison between CG-DCA and baseline methods across various node sizes ($n$), and average in-degrees ($d$) with $N = 5,000$ and $N = 20,000$. For each graph structure, we generate 10 independent data instances using different random seeds and record the average performance metrics.

Table 1 reports the BIC score gap (difference between the achieved BIC score and BIC score of the true graph) and the runtime (with a 3-hour time limit per instance) of CG-DCA and GOBNILP. As demonstrated in Table 1, the CG-DCA method consistently outperforms GOBNILP in terms of average scores across simulated Gaussian datasets. Notably, CG-DCA achieves optimal BIC scores for instances with sparse graph structures ($d = 0.5$). As the number of nodes increases, CG-DCA exhibits a slower growth in runtime compared to GOBNILP while maintaining superior scoring performance. The robustness in both computational efficiency and solution quality with respect to node count ($n$) and graph density ($d$) makes CG-DCA more suitable for larger problems.

In Table 2, we compare four methods with the following metrics on the quality of the solution graphs: (1) precision and recall [46], which measure the proportion of true edges among predicted edges and predicted edges among true edges, respectively; (2) the structural Hamming distance (SHD) [47], which quantifies the dissimilarity between two graphs by counting the required edge additions, deletions, or reversals to make one graph identical to another [3]. Prior to computing these metrics, we convert both predicted and true graphs into their corresponding essential graphs [3], which represent their Markov equivalence classes. The conversion of essential graphs is performed following the methodology described in [38]. We also have results for HC but only present them in Appendix D.2 due to its inferior performance. Notably, CG-DCA achieves better average recall across most instances while maintaining competitive precision and SHD compared to other constraint-based and

Table 3: Performance Comparison of CG-DCA, GOBNILP (GI), (Stable-)PC, and MMHC on Discrete Datasets

| Dataset | Precision | | | | Recall | | | | SHD | | | |
|---|---|---|---|---|---|---|---|---|---|---|---|---|
| | CG-DCA | GI | PC | MMHC | CG-DCA | GI | PC | MMHC | CG-DCA | GI | PC | MMHC |
| LUCAS | 0.77 | **1.00** | 0.83 | 0.33 | 0.83 | **1.00** | 0.83 | 0.33 | 3 | **0** | 2 | 8 |
| INSURANCE | 0.53 | 0.90 | **0.91** | 0.64 | 0.44 | **0.83** | 0.58 | 0.35 | 37 | **11** | 22 | 34 |
| ALARM | 0.46 | **0.87** | 0.83 | 0.47 | 0.50 | **0.89** | 0.76 | 0.33 | 35 | **7** | 11 | 32 |

hybrid approaches. Better performance on recall than on precision and SHD indicates that CG-DCA tends to select denser graphs than others.

### 5.3 Results on Discrete Data

Table 3 compares four methods on discrete datasets LUCAS [23] with $(n, N, d) = (12, 2000, 1)$, INSURANCE [10] with $(n, N, d) = (27, 20000, 3.85)$ and ALARM [7] with $(n, N, d) = (37, 20000, 2.49)$. GOBNILP demonstrates superior performance in all three discrete datasets.

Although CG-DCA is an option for small discrete dataset, its performance degrades with larger networks. This is because the discrete BIC score contains a highly supermodular penalization term $(a_i - 1) \prod_{j \in J} a_j$, making DCA less effective for DS optimization. Thus, development of an improved decomposition strategy for discrete scores (potentially other scores like BDeu [11]) is essential to broaden the applicability of DCA to larger discrete Bayesian networks.

## 6 Conclusion

In this paper, we propose CG-DCA, a method that leverages difference-of-submodular minimization to solve the pricing problem within CG framework for BNSL. Empirical results demonstrate that CG-DCA outperforms state-of-the-art score-based methods on simulated Gaussian datasets with varying node sizes, sample sizes, and graph densities, yielding solutions of high quality.

While CG-DCA is a viable approach for small graphs with discrete data, its scalability to larger graphs remains limited due to the high supermodularity induced by the penalty term in the discrete BIC function. To mitigate this limitation, one could explore alternative decomposition strategies for (potentially different) discrete scoring functions or develop new DS optimization techniques tailored to this specific computational challenge. Furthermore, building upon our efficient pricing heuristic, future research could be focused on developing exact pricing algorithms to enhance both the convergence guarantees and solution optimality of CG, by leveraging exact approaches for submodular [36] and supermodular [39] optimization.

## Acknowledgements

This work was supported by the National Natural Science Foundation of China under Grant 72501249.

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

# A  Definitions

## A.1  The Data Matrix and Scoring Functions

We denote $D$ as the dataset, which is an $N \times n$ matrix where $N$ is the sample size and $n$ is the number of random variables (nodes in Bayesian Network). The entry $D_{ij}$ is the value of $j$-th variable $X_j$ in the $i$-th observation. For discrete labeled data, each label is mapped to a distinct numerical value.

We use the $\ell_0$-penalized log-likelihood function as the scoring function of a DAG, which is

$$\texttt{score}(G; D) := \log(L(G; D)) - \Lambda \cdot k(G)$$

for some $\Lambda \geq 0$. Here, $L(G; D)$ is the likelihood of the graph structure $G$ under data $D$, and $k(G)$ is the number of free parameters to be estimated in the graphical model. A crucial property of this scoring function is that it can be decomposed into node-specific local scores

$$\texttt{score}(G; D) = \sum_{i=1}^{n} \texttt{score}_i(\mathrm{pa}_i(G)),$$

where each local score $\texttt{score}_i(\mathrm{pa}_i(G))$ only depends on a node $i$ and its parent set $\mathrm{pa}_i(G)$.

For continuous data, we assume that variables $\{X_i\}_{i \in V}$ are mean-normalized such that $\mathbb{E}[X_i] = 0$. We also assume that if $J$ is the parent set of node $i$, then $X_i \sim N(\alpha_{iJ}^\top \mathbf{X}_J, \sigma_{i \leftarrow J}^2)$ [48], where $\mathbf{X}_J = \{X_j\}_{j \in J}$. The parameters $\alpha_{iJ}$ and $\sigma_{i \leftarrow J}^2$ are unknown. The likelihood function defining $\texttt{score}_i(J)$ has included these parameters by their maximum likelihood estimator, thus its value only depends on $J = \mathrm{pa}_i(G)$. The resulting local score $\texttt{score}_i(J)$ in this context is

$$\texttt{score}_i(J) = -\frac{N}{2}(1 + \log(2\pi)) - \frac{N}{2}\log(\hat{\sigma}_{i \leftarrow J}^2) - \Lambda|J|,$$

where

$$\hat{\sigma}_{i \leftarrow J}^2 = \min_{\alpha \in \mathbb{R}^{|J|}} \mathbb{E}_{\hat{\mathbb{P}}}\left[(\alpha^\top \mathbf{X}_J - X_i)^2\right]$$

denotes the empirical residual variance (under the empirical distribution $\hat{\mathbb{P}}$) of the linear regression predicting $X_i$ (with $\mathbb{E}[X_i] = 0$) from the features $\mathbf{X}_J$. For simplicity, we will ignore the constant $-\frac{N}{2}(1 + \log(2\pi))$ in our pricing problem optimization framework established based on Proposition 1.

For discrete data, we assume that if $J$ is the parent set of node $i$, then $X_i$ follows a multinomial distribution with parameters depending on the configuration of the parent set values $\mathbf{X}_J := \{X_j\}_{j \in J}$. The local score for discrete data using multinomial likelihood is

$$\texttt{score}_i(J) = \sum_{x_i \in \mathcal{S}_i} \sum_{\mathbf{x}_J \in \mathcal{S}_J} \#(x_i, \mathbf{x}_J) \log\left(\frac{\#(x_i, \mathbf{x}_J)}{\#(\mathbf{x}_J)}\right) - \Lambda(a_i - 1)\prod_{j \in J} a_j,$$

where $\mathcal{S}_i$ and $\mathcal{S}_J$ represent the sets of possible values that $X_i$ and $\mathbf{X}_J$ can take, respectively. The arity (i.e., the number of possible values it can take) of variable $X_j$ is denoted by $a_j$ for $j = 1, \ldots, n$. The count function $\#(x_i, \mathbf{x}_J) = \texttt{Count}(X_i = x_i, \mathbf{X}_J = \mathbf{x}_J)$ counts joint occurrences in the dataset $D$, and $\#(\mathbf{x}_J) = \texttt{Count}(\mathbf{X}_J = \mathbf{x}_J)$ provides the corresponding marginal counts.

## A.2  Submodular Function and Lovász Extension Function

We adopt standard definitions of the submodular set function and the Lovász extension [36].

**Definition 1** (Submodular Set Function). *Let $V$ be a finite ground set. A set function $f : 2^V \to \mathbb{R}$ is submodular if it satisfies the diminishing return property, i.e., for all $A \subseteq B \subseteq V$ and $v \in V \setminus B$,*

$$f(A \cup \{v\}) - f(A) \geq f(B \cup \{v\}) - f(B)$$

*A function is supermodular if $-f$ is submodular, and is modular if it is both submodular and supermodular.*

**Definition 2** (Lovász Extension). *Let $f : 2^V \to \mathbb{R}$ be a set function defined on a ground set $V = \{1, \ldots, d\}$. Given a point $x \in [0, 1]^d$, let $\sigma$ be a permutation of $V$ such that*

$$x_{\sigma(1)} \geq x_{\sigma(2)} \geq \cdots \geq x_{\sigma(d)}.$$

Define the nested subsets $S_k^\sigma = \{\sigma(1), \ldots, \sigma(k)\}$ for $k \in \{1, \ldots, d\}$, and $S_0^\sigma = \emptyset$. The Lovász extension $f^L : [0, 1]^d \to \mathbb{R}$ of the set function $f$ at point $x$ is defined as

$$f^L(x) = \sum_{k=0}^{d} \big(x_{\sigma(k)} - x_{\sigma(k+1)}\big) f(S_k^\sigma),$$

where $x_{\sigma(0)} := 1$ and $x_{\sigma(d+1)} := 0$.

## B   Row Generation and the Separation Problem

The exponential number of cluster constraints can be handled through row generation [15]. This cutting plane approach starts with a restricted set $\hat{\mathcal{C}}$ and sequentially adds constraints violated by the current solution $x^*$. For an integer solution $x^*$, the separation problem identifies violated cluster constraints via cycle detection in the decoded graph from $x^*$ (using, e.g., depth-first search). For a fractional solution of RMLP, the separation problem identifies maximally violated cluster constraints through

$$\max_{C} \left\{ \sum_{i \in C} \sum_{J \cap C \neq \emptyset} x^*_{i \leftarrow J} - |C| \right\},$$

which can be formulated as an *Separation IP* as follows:

$$\max_{y,z} \quad \sum_{i=1}^{n} \sum_{J \in \hat{\mathcal{P}}_i} x^*_{i \leftarrow J} \cdot y_{iJ} - \sum_{i=1}^{n} z_i \tag{5a}$$

$$\text{s.t.} \quad y_{iJ} \leq z_i, \qquad\qquad i = 1, \ldots, n, \quad J \in \hat{\mathcal{P}}_i, \tag{5b}$$

$$y_{iJ} \leq \sum_{i' \in J} z_{i'}, \qquad\quad i = 1, \ldots, n, \quad J \in \hat{\mathcal{P}}_i, \tag{5c}$$

$$\sum_{i=1}^{n} z_i \geq 1, \tag{5d}$$

$$y_{iJ} \in \{0, 1\}, z_i \in \{0, 1\}, \quad i = 1, \ldots, n, \quad J \in \hat{\mathcal{P}}_i. \tag{5e}$$

The optimal solution $(y^*, z^*)$ of Problem (5) defines the new cluster $C^* = \{i : z_i^* = 1\}$ to be added to $\hat{\mathcal{C}}$.

The complete row and column generation algorithm iteratively alternates between column generation for variable selection and row generation for constraint enforcement, dynamically refining both the solution space and constraint set.

## C   Proofs

### C.1   Proof of Proposition 1

*Proof.* Recall that for Gaussian data,

$$\texttt{score}_i(J) = -\frac{N}{2} \log(2\pi + 1) - \frac{N}{2} \log(\hat{\sigma}^2_{i \leftarrow J}) - \Lambda|J|.$$

The conditional variance $\hat{\sigma}^2_{i \leftarrow J}$ satisfies:

$$\begin{aligned}
\hat{\sigma}^2_{i \leftarrow J} &= \min_{\alpha \in \mathbb{R}^{|J|}} \mathbb{E}_{\hat{\mathbb{P}}}[(\alpha^\top \mathbf{X}_J - X_i)^2] \\
&= \min_{\alpha \in \mathbb{R}^{|J|}} \alpha^\top \hat{\Sigma}_{J,J} \alpha - 2\alpha^\top \hat{\Sigma}_{J,i} + \hat{\Sigma}_{i,i} \\
&= \hat{\Sigma}_{i,i} - \hat{\Sigma}_{i,J} \hat{\Sigma}_{J,J}^{-1} \hat{\Sigma}_{J,i} \\
&= \det(\hat{\Sigma}_{J \cup i, J \cup i}) / \det(\hat{\Sigma}_{J,J}),
\end{aligned}$$

where the last equality is due to the Schur complement, $\hat{\mathbb{P}}$ denotes the empirical distribution, with $\hat{\Sigma}_{J,J}, \hat{\Sigma}_{i,J}, \hat{\Sigma}_{J \cup i, J \cup i}$ and $\hat{\Sigma}_{i,i}$ representing the empirical covariance matrices of variable set $\mathbf{X}_J$, cross-covariance between $X_i$ and $\mathbf{X}_J$, joint covariance of $\mathbf{X}_J \cup X_i$ and variance of $X_i$, respectively.

The logarithmic transformation yields

$$\log(\hat{\sigma}_{i \leftarrow J}^2) = \log \det(\hat{\Sigma}_{J \cup i, J \cup i}) - \log \det(\hat{\Sigma}_{J,J}),$$

where both $\log \det(\hat{\Sigma}_{J,J})$ and $\log \det(\hat{\Sigma}_{J \cup i, J \cup i})$ are submodular functions of set $J$ [34]. Function $\Lambda |J| + \lambda_i^*$ is modular in $J$.

The remaining term involving $J$, $\sum_{\substack{C \in \hat{\mathcal{C}}: \\ i \in C, \, J \cap C \neq \emptyset}} \lambda_C^*$, in the pricing objective is also submodular in $J$.
This follows from the property that for any $J_1 \subseteq J_2$ and $j \notin J_2$,

$$\sum_{C \in \hat{\mathcal{C}}: \, i \in C, \, J_1 \cap C = \emptyset} \lambda_C^* \geq \sum_{C \in \hat{\mathcal{C}}: \, i \in C, \, J_2 \cap C = \emptyset} \lambda_C^*.$$

Given that $\lambda_C^* \geq 0$, the above inequality holds since $J_1 \subseteq J_2$ implies that $C \cap J_2 = \emptyset$ necessitates $C \cap J_1 = \emptyset$.

Consequently, the pricing objective to be minimized can be expressed as the following DS function:

$$z(J; \lambda^*) = \frac{N}{2} \log \det(\hat{\Sigma}_{J \cup i, J \cup i}) + \Lambda|J| + \underbrace{\sum_{\substack{C \in \hat{\mathcal{C}}: \\ i \in C, \, J \cap C \neq \emptyset}} \lambda_C^* + \lambda_i^*}_{\text{submodular}} \underbrace{- \frac{N}{2} \log \det(\hat{\Sigma}_{J,J})}_{\text{submodular}} + \frac{N}{2} \log(2\pi + 1).$$

$\square$

### C.2 Proof of Proposition 2

*Proof.* Recall that for multinomial data,

$$\mathtt{score}_i(J) = \sum_{x_i \in \mathcal{S}_i} \sum_{\mathbf{x}_J \in \mathcal{S}_J} \#(x_i, \mathbf{x}_J) \log \left( \frac{\#(x_i, \mathbf{x}_J)}{\#(\mathbf{x}_J)} \right) - \Lambda(a_i - 1) \prod_{j \in J} a_j.$$

We reformulate the first term as follows:

$$\sum_{x_i \in S_i} \sum_{\mathbf{x}_J \in S_J} \#(x_i, \mathbf{x}_J) \log \left( \frac{\#(x_i, \mathbf{x}_J)}{\#(\mathbf{x}_J)} \right)$$

$$= \sum_{x_i \in S_i} \sum_{\mathbf{x}_J \in S_J} \#(x_i, \mathbf{x}_J) \log(\#(x_i, \mathbf{x}_J)) - \sum_{\mathbf{x}_J \in S_J} \#(\mathbf{x}_J) \log(\#(\mathbf{x}_J))$$

$$= N \left[ \sum_{x_i \in S_i} \sum_{\mathbf{x}_J \in S_J} \hat{\mathbb{P}}(X_i = x_i, \mathbf{X}_J = \mathbf{x}_J) \log \hat{\mathbb{P}}(X_i = x_i, \mathbf{X}_J = \mathbf{x}_J) - \sum_{\mathbf{x}_J \in S_J} \hat{\mathbb{P}}(\mathbf{X}_J = \mathbf{x}_J) \log \hat{\mathbb{P}}(\mathbf{X}_J = \mathbf{x}_J) \right]$$

$$= N(-H(J \cup \{i\}) + H(J)),$$

where $N$ is the sample size. The functions

$$H(J) := - \sum_{\mathbf{x}_J \in S_J} \hat{\mathbb{P}}(\mathbf{X}_J = \mathbf{x}_J) \log \hat{\mathbb{P}}(\mathbf{X}_J = \mathbf{x}_J)$$

and

$$H(J \cup \{i\}) := - \sum_{x_i \in S_i} \sum_{\mathbf{x}_J \in S_J} \hat{\mathbb{P}}(X_i = x_i, \mathbf{X}_J = \mathbf{x}_J) \log \hat{\mathbb{P}}(X_i = x_i, \mathbf{X}_J = \mathbf{x}_J)$$

denote the entropy functions, which are known to be submodular [24].

By the proof of Proposition 1, $\sum_{\substack{C \in \hat{\mathcal{C}}: \\ i \in C, \, J \cap C \neq \emptyset}} \lambda_C^*$ is submodular in $J$. It is also easy to verify that $(a_i - 1) \prod_{j \in J} a_j$ is supermodular in $J$ as $a_j \geq 1$ for $j \in J \cup \{i\}$.

Therefore, the pricing objective can be expressed as the following DS function:

$$z(J; \lambda^*) = \underbrace{N \cdot H(J \cup \{i\}) + \sum_{\substack{C: C \in \mathcal{C} \\ i \in C \\ C \cap J \neq \emptyset}} \lambda_C^* + \lambda_i^*}_{\text{submodular}} - \underbrace{\left( N \cdot H(J) - \Lambda(a_i - 1) \prod_{j \in J} a_j \right)}_{\text{submodular}}.$$

$\square$

# D   Supplementary Numerical Results

## D.1   Comparison of DCA initialization methods

To evaluate the sensitivity of DCA initialization, we compare the BIC scores and time costs across three initialization methods (warm-start, random, and hybrid). The results are summarized in Table 4 and 5:

Table 4: BIC Score Comparison of the Three Initialization Approaches for DCA

| $(n, N, d)$ | hybrid | random | warmstart |
|---|---|---|---|
| (15, 5000,0.5) | **-101818.64** | **-101818.64** | -101819.58 |
| (15, 5000,1.0) | -101954.63 | **-101939.75** | -101975.70 |
| (15, 5000,1.5) | -101411.14 | **-101376.34** | -101509.59 |
| (15, 5000,2.0) | -101646.26 | **-101556.54** | -101608.25 |
| (15, 20000,0.5) | **-406345.25** | **-406345.25** | -406346.23 |
| (15, 20000,1.0) | -410415.68 | -410415.68 | **-410382.11** |
| (15, 20000,1.5) | **-405344.24** | -405487.19 | -405599.33 |
| (15, 20000,2.0) | -400909.42 | **-400861.51** | -401710.97 |
| (20, 5000,0.5) | **-136605.28** | **-136605.28** | -136734.74 |
| (20, 5000,1.0) | -137222.13 | **-137212.97** | -137363.27 |
| (20, 5000,1.5) | -136624.81 | **-136577.03** | -136957.87 |
| (20, 5000,2.0) | -136849.69 | **-136610.04** | -136750.44 |
| (20, 20000,0.5) | **-535047.92** | **-535047.92** | -535183.20 |
| (20, 20000,1.0) | **-541850.16** | -541887.02 | -542168.27 |
| (20, 20000,1.5) | **-538682.19** | -538693.99 | -539863.90 |
| (20, 20000,2.0) | -535301.38 | **-534115.87** | -534939.03 |
| (25, 5000,0.5) | **-170751.53** | **-170751.53** | -170751.74 |
| (25, 5000,1.0) | **-170279.79** | -170296.43 | -170302.01 |
| (25, 5000,1.5) | -171308.77 | **-171157.64** | -171289.38 |
| (25, 5000,2.0) | -171333.89 | -171105.48 | **-170760.08** |
| (25, 20000,0.5) | -677813.87 | -677813.87 | **-677812.65** |
| (25, 20000,1.0) | -683475.44 | **-683189.30** | -683826.26 |
| (25, 20000,1.5) | **-676190.15** | -676267.45 | -676649.76 |
| (25, 20000,2.0) | -689041.33 | -688690.44 | **-687147.17** |

*Each value is the average over ten independent instances.

Table 5: Runtime Comparison of the Three Initialization Approaches for DCA

| $(n, N, d)$ | hybrid (s) | random (s) | warmstart (s) |
|---|---|---|---|
| (15, 5000, 0.5) | 4.06 | 4.04 | **1.17** |
| (15, 5000, 1.0) | 25.76 | 29.40 | **4.49** |
| (15, 5000, 1.5) | 76.72 | 87.43 | **10.00** |
| (15, 5000, 2.0) | 170.39 | 295.25 | **23.74** |
| (15, 20000, 0.5) | 10.17 | 10.94 | **3.51** |
| (15, 20000, 1.0) | 36.71 | 36.57 | **10.63** |
| (15, 20000, 1.5) | 104.25 | 109.73 | **23.95** |
| (15, 20000, 2.0) | 229.80 | 245.11 | **41.94** |
| (20, 5000, 0.5) | 9.53 | 9.60 | **2.42** |
| (20, 5000, 1.0) | 84.34 | 90.87 | **12.51** |
| (20, 5000, 1.5) | 312.67 | 430.37 | **48.78** |
| (20, 5000, 2.0) | 1039.53 | 1770.71 | **95.36** |
| (20, 20000, 0.5) | 18.42 | 18.54 | **8.71** |
| (20, 20000, 1.0) | 133.69 | 134.79 | **34.84** |
| (20, 20000, 1.5) | **425.17** | 504.29 | 444.37 |
| (20, 20000, 2.0) | 997.99 | 1710.91 | **211.44** |
| (25, 5000, 0.5) | 21.97 | 22.35 | **3.87** |
| (25, 5000, 1.0) | 261.84 | 304.12 | **33.59** |
| (25, 5000, 1.5) | 949.90 | 2434.03 | **131.69** |
| (25, 5000, 2.0) | 3054.63 | 6037.58 | **224.73** |
| (25, 20000, 0.5) | 50.22 | 53.13 | **12.48** |
| (25, 20000, 1.0) | 404.43 | 642.87 | **78.21** |
| (25, 20000, 1.5) | 1372.59 | 3688.33 | **227.15** |
| (25, 20000, 2.0) | 3189.98 | 6674.62 | **366.46** |

*Each value is the average over ten independent instances.

From the tables, we observe that

- Random initialization yields highest (best among three) BIC scores in 15 out of 24 instances among three methods but requires the most time in 22 out of 24 instances.

- Warm-start initialization achieves the lowest (worst among three) BIC scores in 16 out of 24 instances and is the most time-efficient in 23 out of 24 instances.

- The hybrid approach strikes a balance between optimality and computational efficiency, since it focuses on local refinement around the current best pattern (through warm-start initialization) while building upon the foundation of global exploration (through random initialization) at the early stage.

## D.2 Comparison of CG-DCA with Baselines

Table 6 presents the comprehensive experimental results including the HC method, with $N \in \{5000, 20000\}$ for Gaussian datasets. For the three discrete datasets, Table 7 provides the score and time comparison for CG-DCA and GOBNILP, while Table 8 provides the graph comparisons of all baselines. The runtime performance of all baseline methods (implemented in R) on all datasets is summarized in Table 9 and 10.

Table 6: Performance of CG-DCA, GOBNILP (GI), HC, (Stable-)PC, and MMHC on Gaussian Datasets

| $n, N, d$ | Precision | | | | | Recall | | | | | SHD | | | | |
|---|---|---|---|---|---|---|---|---|---|---|---|---|---|---|---|
| | CG-DCA | GI | HC | PC | MMHC | CG-DCA | GI | HC | PC | MMHC | CG-DCA | GI | HC | PC | MMHC |
| (15, 5000, 0.5) | **0.78** | 0.66 | 0.36 | 0.59 | 0.50 | **0.80** | 0.73 | 0.47 | 0.62 | 0.51 | **1.70** | 2.90 | **6.20** | 3.00 | 3.50 |
| (15, 5000, 1.0) | 0.70 | 0.67 | 0.41 | **0.72** | 0.59 | **0.78** | 0.72 | 0.52 | 0.67 | 0.54 | 6.20 | 5.00 | 11.90 | **4.80** | 6.70 |
| (15, 5000, 1.5) | 0.65 | **0.73** | 0.42 | 0.61 | 0.56 | 0.77 | **0.80** | 0.57 | 0.54 | 0.48 | 11.80 | **7.40** | 19.50 | 11.50 | 12.20 |
| (15, 5000, 2.0) | 0.55 | **0.58** | 0.34 | 0.54 | 0.52 | **0.72** | 0.65 | 0.57 | 0.39 | 0.36 | 21.90 | **16.90** | 35.30 | 20.90 | 20.40 |
| (15, 20000, 0.5) | **0.71** | 0.69 | 0.50 | 0.68 | 0.53 | **0.74** | 0.73 | 0.52 | 0.68 | 0.53 | 2.20 | 2.20 | 3.70 | **1.90** | 3.00 |
| (15, 20000, 1.0) | 0.76 | **0.78** | 0.49 | 0.71 | 0.61 | 0.78 | **0.81** | 0.61 | 0.69 | 0.59 | 4.10 | **4.00** | 9.60 | 4.70 | 6.20 |
| (15, 20000, 1.5) | **0.76** | 0.73 | 0.47 | 0.72 | 0.66 | **0.85** | 0.79 | 0.61 | 0.66 | 0.57 | **6.90** | 9.40 | 18.30 | 8.60 | 10.00 |
| (15, 20000, 2.0) | **0.61** | 0.53 | 0.38 | 0.53 | 0.55 | **0.75** | 0.60 | 0.62 | 0.43 | 0.39 | 20.60 | 19.90 | 33.20 | 20.01 | **19.80** |
| (20, 5000, 0.5) | 0.74 | 0.71 | 0.48 | **0.76** | 0.59 | **0.80** | 0.75 | 0.56 | 0.79 | 0.60 | 3.40 | 3.50 | 6.80 | **2.60** | 4.40 |
| (20, 5000, 1.0) | 0.71 | 0.69 | 0.49 | **0.75** | 0.65 | 0.76 | **0.77** | 0.63 | 0.71 | 0.59 | **3.70** | 7.50 | 13.20 | 5.90 | 7.90 |
| (20, 5000, 1.5) | 0.50 | 0.61 | 0.41 | 0.62 | **0.65** | **0.71** | 0.71 | 0.64 | 0.50 | 0.49 | 28.20 | **16.80** | 30.60 | 17.90 | **16.80** |
| (20, 5000, 2.0) | 0.44 | 0.46 | 0.41 | 0.54 | **0.55** | **0.66** | 0.53 | 0.67 | 0.34 | 0.31 | 36.40 | 33.60 | 49.80 | 31.50 | **31.10** |
| (20, 20000, 0.5) | **0.82** | 0.80 | 0.54 | 0.73 | 0.58 | **0.83** | 0.82 | 0.62 | 0.76 | 0.59 | **2.20** | 2.60 | 5.70 | 3.00 | 4.60 |
| (20, 20000, 1.0) | 0.61 | 0.73 | 0.56 | **0.79** | 0.68 | 0.70 | **0.79** | 0.67 | 0.77 | 0.65 | 8.30 | 6.90 | 10.80 | **4.90** | 7.10 |
| (20, 20000, 1.5) | **0.65** | 0.58 | 0.40 | 0.64 | 0.63 | **0.81** | 0.68 | 0.64 | 0.56 | 0.50 | 16.70 | 18.20 | 32.80 | **16.00** | 16.50 |
| (20, 20000, 2.0) | 0.52 | 0.48 | 0.41 | 0.51 | **0.56** | **0.73** | 0.55 | 0.67 | 0.37 | 0.33 | 33.00 | 32.40 | 46.00 | 31.30 | **29.30** |
| (25, 5000, 0.5) | 0.82 | 0.61 | 0.60 | **0.84** | 0.69 | **0.84** | 0.67 | 0.67 | 0.84 | 0.69 | **2.70** | 5.90 | 6.70 | **2.70** | 4.40 |
| (25, 5000, 1.0) | **0.81** | 0.69 | 0.57 | 0.80 | 0.74 | **0.86** | 0.76 | 0.71 | 0.75 | 0.69 | **6.70** | 11.10 | 16.30 | 7.40 | 8.70 |
| (25, 5000, 1.5) | 0.66 | 0.57 | 0.44 | 0.70 | **0.71** | **0.75** | 0.68 | 0.65 | 0.58 | 0.56 | **13.60** | 22.10 | 39.50 | 18.50 | 18.00 |
| (25, 5000, 2.0) | 0.54 | 0.47 | 0.38 | 0.53 | **0.59** | **0.73** | 0.54 | 0.67 | 0.35 | 0.35 | 45.70 | 40.30 | 63.70 | 38.70 | **36.50** |
| (25, 20000, 0.5) | 0.76 | 0.80 | 0.59 | **0.81** | 0.70 | 0.80 | **0.84** | 0.66 | 0.83 | 0.70 | 3.70 | **2.80** | 7.30 | **2.80** | 4.30 |
| (25, 20000, 1.0) | 0.76 | 0.71 | 0.51 | **0.81** | 0.68 | **0.86** | 0.76 | 0.67 | 0.78 | 0.64 | 9.10 | 9.90 | 17.10 | **6.70** | 9.90 |
| (25, 20000, 1.5) | 0.57 | 0.53 | 0.40 | **0.73** | 0.63 | **0.67** | 0.66 | 0.62 | 0.63 | 0.51 | **14.10** | 25.00 | 41.90 | 16.50 | 19.50 |
| (25, 20000, 2.0) | 0.54 | 0.48 | 0.35 | **0.61** | 0.59 | **0.66** | 0.56 | 0.64 | 0.46 | 0.40 | **23.90** | 39.10 | 77.00 | 33.80 | 34.10 |

*Each value is the average over ten independent instances.

Table 7: Score and Time Comparison of CG-DCA and GOBNILP on Discrete Datasets

| Dataset | BIC Score Gap (%) | | Time (seconds) | |
|---|---|---|---|---|
| | CG-DCA | GOBNILP | CG-DCA | GOBNILP |
| LUCAS | 0.03 | **0.00** | 11.59 | **7.99** |
| INSURANCE | 0.75 | **0.00** | 1892.36 | **74.39** |
| ALARM | 2.85 | **0.00** | 4696.19 | **153.49** |

Table 8: Performance Comparison of CG-DCA, GOBNILP (GI), HC, PC, and MMHC on Discrete Datasets

| Dataset | Precision | | | | | Recall | | | | | SHD | | | | |
|---|---|---|---|---|---|---|---|---|---|---|---|---|---|---|---|
| | CG-DCA | GI | HC | PC | MMHC | CG-DCA | GI | HC | PC | MMHC | CG-DCA | GI | HC | PC | MMHC |
| LUCAS | 0.77 | **1.00** | 0.29 | 0.83 | 0.33 | 0.83 | **1.00** | 0.33 | 0.83 | 0.33 | 3 | **0** | 10 | 2 | 8 |
| INSURANCE | 0.53 | 0.90 | 0.52 | **0.91** | 0.64 | 0.44 | **0.83** | 0.50 | 0.58 | 0.35 | 37 | **11** | 38 | 22 | 34 |
| ALARM | 0.46 | **0.87** | 0.42 | 0.83 | 0.47 | 0.50 | **0.89** | 0.48 | 0.76 | 0.33 | 35 | **7** | 34 | 11 | 32 |

Table 9: Time Comparison of Stable PC, HC, and MMHC Algorithms on Gaussian Datasets

| $(n, N, d)$ | Stable PC (s) | HC (s) | MMHC (s) |
|---|---|---|---|
| (15,  5000, 0.5) | 0.038 | 0.048 | 0.048 |
| (15,  5000, 1.0) | 0.035 | 0.054 | 0.047 |
| (15,  5000, 1.5) | 0.054 | 0.101 | 0.062 |
| (15,  5000, 2.0) | 0.095 | 0.098 | 0.092 |
| (15, 20000, 0.5) | 0.097 | 0.154 | 0.126 |
| (15, 20000, 1.0) | 0.121 | 0.196 | 0.152 |
| (15, 20000, 1.5) | 0.188 | 0.263 | 0.326 |
| (15, 20000, 2.0) | 0.306 | 0.420 | 0.889 |
| (20,  5000, 0.5) | 0.050 | 0.066 | 0.090 |
| (20,  5000, 1.0) | 0.105 | 0.159 | 0.112 |
| (20,  5000, 1.5) | 0.180 | 0.309 | 0.291 |
| (20,  5000, 2.0) | 0.266 | 0.342 | 0.425 |
| (20, 20000, 0.5) | 0.146 | 0.255 | 0.185 |
| (20, 20000, 1.0) | 0.279 | 0.481 | 0.438 |
| (20, 20000, 1.5) | 0.660 | 0.950 | 1.366 |
| (20, 20000, 2.0) | 1.327 | 2.331 | 3.634 |
| (25,  5000, 0.5) | 0.083 | 0.099 | 0.071 |
| (25,  5000, 1.0) | 0.095 | 0.133 | 0.098 |
| (25,  5000, 1.5) | 0.227 | 0.348 | 0.218 |
| (25,  5000, 2.0) | 0.273 | 0.403 | 0.351 |
| (25, 20000, 0.5) | 0.200 | 0.407 | 0.235 |
| (25, 20000, 1.0) | 0.288 | 0.637 | 0.583 |
| (25, 20000, 1.5) | 0.725 | 1.568 | 2.612 |
| (25, 20000, 2.0) | 1.158 | 2.087 | 2.940 |

*Each value is the average over ten independent instances.

Table 10: Time Comparison of Stable PC, HC, and MMHC Algorithms on Discrete Datasets

| Dataset | Stable PC (s) | HC (s) | MMHC (s) |
|---|---|---|---|
| LUCAS | 0.08 | 0.08 | 0.08 |
| INSURANCE | 0.65 | 0.56 | 0.45 |
| ALARM | 0.61 | 0.75 | 0.53 |

