# OpenReview forum: "Inexact Column Generation for Bayesian Network Structure Learning via Difference-of-Submodular Optimization"
_NeurIPS.cc/2025/Conference — NeurIPS 2025 poster_

### Official Review · Reviewer_VSAx · 2025-06-25

**Clarity:** 4
**Significance:** 3
**Originality:** 2
**Rating:** 5
**Confidence:** 3

**Summary:**

The authors propose a new algorithm for Bayesian Network Structure Learning (BNSL). The proposed method is based on the score-based Integer Programming (IP) approach and solves the formulated IP using row and column generation methods. The column generation method requires solving a set optimization problem called the pricing problem, which can be expressed as the difference of two submodular functions, and the problem can be solved efficiently by applying the Discrete Convex Algorithm (DCA) with Lovász extension. Experiments on discrete and continuous synthetic and real data confirm that the proposed method outperforms existing methods.

**Questions:**

- What part of the title refers to “Inexact”? Is the information “Inexact” important enough to include in the title? This part should be explained more emphatically in the text.
- I did not understand the explanation in lines 145-148, where the output of Algorithm 1 is always an integer solution. I would appreciate a more detailed explanation. In particular, the part after “all extreme point solutions ...” should be explained.

**Ethical Concerns:**

["NO or VERY MINOR ethics concerns only"]

**Final Justification:**

My concerns regarding
* the global optimality of the DCA,
* the choice of initial values, and
* the other questions I raised

have all been satisfactorily resolved. Consequently, I have increased my assigned score.

**Limitations:**

- As the authors mention in the Conclusion, scalability is limited.
- Theoretical guarantees regarding how well DCA can solve column generation problems are difficult to obtain.

**Quality:**

3

**Strengths And Weaknesses:**

# Strengths
- The proposed method, although heuristic, is practical and sound for the important problem of BNSL.
- The superiority of the proposed method is experimentally confirmed on various data sets.
- The paper is well-structured and very easy to read.

# Weaknesses
- In the machine learning field, the procedure of solving NP-hard problems by formulating them as Difference of Convex or Difference of Submodular is often used. Therefore, I feel that the proposed method, while effective for the problem, is not that novel.
- DCA is not necessarily a globally optimal algorithm, and there are no theoretical guarantees regarding the quality of the solution.
- Section 4.1 describes the importance of the initial value setting, but does not provide experimental results for this part, so there is a lack of information on what guidelines users of this algorithm should use for their initial value strategy.

---

> ### Author Rebuttal · Authors · 2025-07-30
>
> We sincerely appreciate the reviewers' insightful comments. Below we provide our response:
>
> **1. Novelty of the Difference-of-Submodular Formulation**\
> Our work introduces a key methodological innovation through a novel decomposition of the pricing objective. To the best of our knowledge, this represents the first successful application of difference-of-submodular optimization techniques to the Bayesian Network Structure Learning (BNSL) problem.
>
> **2. Global Performance of Our Algorithm**\
> The Difference-of-Convex Algorithm (DCA) serves as an efficient heuristic for the pricing subproblem, with its practical effectiveness empirically validated through our numerical study. Our implementation strategically employs DCA in column generation subproblems to rapidly produce high-quality columns. As is common in vehicle routing [1,2] (and pointed out by Reviewer krir), to turn our approach into an exact solution approach for BNSL, one may apply inexact column generation first to quickly generate good columns and then apply exact column generation to conclude that there is no need for additional columns. We believe our results empirically show that DCA serves this purpose, and our framework can be potentially applied to develop such an exact approach.
>
> **3. Initialization Strategies for DCA**\
> We apologize for the previous omission of the comparison. We conducted some addtional experiments. The tables below present a comparison of BIC scores and computational time obtained from our CG-DCA framework by applying the three initialization methods we mentioned in our manuscript. These additional results will be added into Appendix.
>
> | n, N, d      | hybrid       | random       | warmstart    |
> |--------------|--------------|--------------|--------------|
> | 15, 5000, 0.5 | -101818.64   | -101818.64   | -101819.58   |
> | 15, 5000, 1   | -101954.63   | -101939.75   | -101975.70   |
> | 15, 5000, 1.5 | -101411.14   | -101376.34   | -101509.59   |
> | 15, 5000, 2   | -101646.26   | -101556.54   | -101608.25   |
> | 15, 20000, 0.5 | -406345.25   | -406345.25   | -406346.23   |
> | 15, 20000, 1  | -410415.68   | -410415.68   | -410382.11   |
> | 15, 20000, 1.5 | -405344.24   | -405487.19   | -405599.33   |
> | 15, 20000, 2  | -400909.42   | -400861.51   | -401710.97   |
> | 20, 5000, 0.5 | -136605.28   | -136605.28   | -136734.74   |
> | 20, 5000, 1   | -137222.13   | -137212.97   | -137363.27   |
> | 20, 5000, 1.5 | -136624.81   | -136577.03   | -136957.87   |
> | 20, 5000, 2   | -136849.69   | -136610.04   | -136750.44   |
> | 20, 20000, 0.5 | -535047.92   | -535047.92   | -535183.20   |
> | 20, 20000, 1  | -541850.16   | -541887.02   | -542168.27   |
> | 20, 20000, 1.5 | -538682.19   | -538693.99   | -539863.90   |
> | 20, 20000, 2  | -535301.38   | -534115.87   | -534939.03   |
> | 25, 5000, 0.5 | -170751.53   | -170751.53   | -170751.74   |
> | 25, 5000, 1   | -170279.79   | -170296.43   | -170302.01   |
> | 25, 5000, 1.5 | -171308.77   | -171157.64   | -171289.38   |
> | 25, 5000, 2   | -171333.89   | -171105.48   | -170760.08  |
> | 25, 20000, 0.5 | -677813.87   | -677813.87   | -677812.65   |
> | 25, 20000, 1  | -683475.44   | -683189.30   | -683826.26   |
> | 25, 20000, 1.5 | -676190.15   | -676267.45   | -676649.76   |
> | 25, 20000, 2  | -689041.33   | -688690.44   | -687147.17  |
> | **Average**   | **-312,582.26** | **-312,553.87** | **-312,712.62** |
>
> | n, N, d      | hybrid    | random    | warmstart |
> |--------------|--------------|--------------|--------------|
> | 15, 5000, 0.5 | 4.06s     | 4.04s     | 1.17s     |
> | 15, 5000, 1   | 25.76s    | 29.40s    | 4.49s     |
> | 15, 5000, 1.5 | 76.72s    | 87.43s    | 10.00s    |
> | 15, 5000, 2   | 170.39s   | 295.25s   | 23.74s    |
> | 15, 20000, 0.5 | 10.17s    | 10.94s    | 3.51s     |
> | 15, 20000, 1  | 36.71s    | 36.57s    | 10.63s    |
> | 15, 20000, 1.5 | 104.25s   | 109.73s   | 23.95s    |
> | 15, 20000, 2  | 229.80s   | 245.11s   | 41.94s    |
> | 20, 5000, 0.5 | 9.53s     | 9.60s     | 2.42s     |
> | 20, 5000, 1   | 84.34s    | 90.87s    | 12.51s    |
> | 20, 5000, 1.5 | 312.67s   | 430.37s   | 48.78s    |
> | 20, 5000, 2   | 1039.53s  | 1770.71s  | 95.36s    |
> | 20, 20000, 0.5 | 18.42s    | 18.54s    | 8.71s     |
> | 20, 20000, 1  | 133.69s   | 134.79s   | 34.84s    |
> | 20, 20000, 1.5 | 425.17s   | 504.29s   | 444.37s   |
> | 20, 20000, 2  | 997.99s   | 1710.91s  | 211.44s   |
> | 25, 5000, 0.5 | 21.97s    | 22.35s    | 3.87s     |
> | 25, 5000, 1   | 261.84s   | 304.12s   | 33.59s    |
> | 25, 5000, 1.5 | 949.90s   | 2434.03s  | 131.69s   |
> | 25, 5000, 2   | 3054.63s  | 6037.58s  | 224.73s  |
> | 25, 20000, 0.5 | 50.22s    | 53.13s    | 12.48s    |
> | 25, 20000, 1  | 404.43s   | 642.87s   | 78.21s    |
> | 25, 20000, 1.5 | 1372.59s  | 3688.33s  | 227.15s   |
> | 25, 20000, 2  | 3189.98s  | 6674.62s  | 366.46s  |
> | **Average**   | **495.62s** | **897.93s** | **88.71s** |
>
> From the tables, we can observe that:
> - Random initialization yields highest (best amony three)
> BIC scores in 11 out of 24 instances among three methods but requires the most time in 22 out of 24 instances.
> - Warm-start initialization achieves the lowest (worst among three) BIC scores in 16 out of 24 instances and is the most time-efficient in 23 out of 24 instances.
> - The hybrid approach strikes a balance between optimality and computational efficiency, since it focuses on local refinement around the current best pattern (through warm-start initialization) while building upon the foundation of global exploration (through random initialization) at the early stage.
>
> **4. Clarification of "Inexact" in Title**\
> The term "inexact" in our title specifically refers to our heuristic approach for solving the column generation subproblems (pricing problems) approximately during the optimization process. We will make sure that this is explicitly clarified in the published version if accepted.
>
> **5. Integral Extreme Point Solutions**\
> Note that the epigraph of the Lovász extension of a submodular function $h$ charaterizes the convex envelope of (the standard binarization of) the set function $h$ [3]. Since $g^L(\cdot)$ is the Lovász extension of the submodular function $g$,  the expression $g^L(\cdot) - \langle y^t, \cdot \rangle$ is itself the Lovász extension of a submodular function (specifically, $g$ plus a modular function). Consequently, all extreme points of the epigraph of $g^L(\cdot)-\langle y^t,\cdot\rangle$ has binary $x$ components. Therefore, if we optimize $g^L(\cdot)-\langle y^t,\cdot\rangle$ exactly using the cutting plane method in Line 5 of Algorithm 1 and always retrieve an extreme point optimal solution, the solution $x^{t+1}$ would be binary/integral (subject to numerical errors). We will clarify this in an updated version of our manuscript.
>
> We appreciate the opportunity to clarify these points and remain open to further discussion or suggestions.
>
> ## References
> [1] Pessoa, Artur, et al. "A generic exact solver for vehicle routing and related problems." *Mathematical Programming* 183.1 (2020): 483-523.\
> [2] You, Zhengzhong, and Yu Yang. "RouteOpt: An Open-Source Modular Exact Solver for Vehicle Routing Problems." *Available at SSRN* 5314242 (2025).\
> [3] Lovász, László. "Submodular functions and convexity." *Mathematical Programming The State of the Art: Bonn 1982*. Berlin, Heidelberg: Springer Berlin Heidelberg, 1983. 235-257.

---

> > ### Comment · Reviewer_VSAx · 2025-08-05
> >
> > I have reviewed your rebuttal, and since it has fully addressed all of my concerns, I will raise my score.

---

> > > ### Author Response · Authors · 2025-08-05
> > >
> > > Thank you. We appreciate your time reviewing our rebuttal and the positive feedback.

---

### Official Review · Reviewer_6rfE · 2025-07-02

**Clarity:** 3
**Significance:** 2
**Originality:** 2
**Rating:** 5
**Confidence:** 3

**Summary:**

The paper considers the Bayesian network structure learning (BNSL) problem and adopts an existing score-based integer programming (IP) formulation with exponentially many variables and constraints. Row and column generation are standard techniques for handling large IPs. Existing exact score-based IP methods [16, 18] employ these techniques, but solving the pricing problem for column generation (CG) exactly is expensive, which limits their scalability.  This paper builds on [18], but instead of solving the pricing problem exactly, the authors reformulate it as a difference of submodular (DS) function minimization problem for $\ell_0$-penalized likelihood scores, then apply the Difference of Convex Algorithm (DCA) which  was shown to be an effective heuristic in prior work. Empirical results comparing their method to various baselines on continuous and discrete data are provided.

**Questions:**

- Why are baselines not all tested on the same datasets and with all the performance metrics considered (BIC score gap, time, Precision, Recall and SHD)?
- In Section 5.1, it's not clear how are larger instances generated?
- Which implementation of GOBNILP do you use, the python one? It's good to specify that.
- How do you break ties when computing the subgradient  of the Lovász extension $f^L$ on line 4 of Algorithm 1? Prior work showed that performance of DCA [22] / SubSup [30] is sensitive to this choice.
- In [22], a variant of DCA which checks for local minimality at convergence and restarts the algorithm from the best neighboring point if  not satisfied (Algorithm 2 therein) is shown to converge to a local minimum. This could improve performance. Did you try that?
- Can you clarify why the supermodular penalization term in the discrete BIC score makes DCA less effective?
- Include a discussion of the computational complexity of the overall proposed algorithm (Algorithm 2)



Other suggestions:
- I suggest trying to initialize DCA from the solution of the pricing problem from the previous iteration in the for loop on line 5. Also initializing with zero at the first iteration might help avoid the dense solutions observed empirically.
- Provide error bars in results tables
- Discuss pros & cons of using score-based IP approach over other approaches to better motivate the paper
- Provide better review of DS minimization in the main text, e.g., provide references to stated results (some are missing such as the form of the subgradient), explain how to obtain an integer solution.
- In Section 5, it's helpful to clarify what type of methods are the baselines compared with (score-based, constrained-base, hybrid, exact or heuristics, IP, etc).

**Ethical Concerns:**

["NO or VERY MINOR ethics concerns only"]

**Final Justification:**

The authors addressed most of my concerns, and ones raised by other reviewers. The proposed method still have important limitations (high computational complexity, degraded performance on discrete datasets) but is still a good contribution towards better scoring-based methods for the BNSL problem.

**Limitations:**

The authors discuss clearly the limitation of their method on discrete data.
But they do not discuss the societal impact of their work, nor do they explain why their work has no societal impact (as requested in the checklist).

**Paper Formatting Concerns:**

No concerns

**Quality:**

2

**Strengths And Weaknesses:**

Strengths:
- The approach of using DS minimization for solving the pricing problem is novel
- The theoretical results formulating the pricing problem as a DS minimization problem for $\ell_0$-penalized Gaussian and multinomial likelihood scores are correct
- The paper is written clearly
- The proposed method seems to outperform score-based baselines (GOBNILP, CG-MINLP, HC) on synthetic Gaussian data and somewhat comparable to constraint-based (stable-PC) and hybrid (MMHC) baselines.
- Code is provided in supplementary material

Weaknesses:
- Experimental results are not well supported: different baselines are compared on different datasets and with different metrics, and some experimental details are missing (see questions for details).
- The proposed method is computationally expensive, and does not work well on discrete datasets
- The proposed method performs worse than the constraint-based (stable-PC) and hybrid (MMHC) baselines in some settings

---

> ### Author Rebuttal · Authors · 2025-07-30
>
> We sincerely appreciate the reviewer's detailed feedback and thoughtful suggestions. Below we provide the responses.
>
> **1. Experimental Results Presentation**\
> Regarding the experimental results presentation, we acknowledge the confusion and apologize for any lack of clarity. All baseline methods were evaluated on identical datasets using consistent performance metrics, with the exception of runtime comparisons for non-IP-based methods due to their R implementation versus our Python framework. Tables 1 and 2 in our paper share the same datasets, while Tables 2 and 3 employ identical metrics. For non-score-based methods (stable-PC and MMHC), we omitted score reporting as it wouldn't provide meaningful comparison. Results for $N=20,000$ samples are provided in the appendix, and we will additionally include the previously omitted table comparing scores and runtime for $N=20,000$ continuous and discrete data to ensure comprehensive reporting.
>
> **2. Computational Cost**\
> We fully recognize that our proposed approach incurs higher computational costs compared to non-IP methods. However, our primary objective is to develop a truly exact score-based framework where the current implementation serves as an efficient subroutine for column generation. In the meanwhile, we have achieved some efficiency improvement though some implementation enhancements, including:
> - Subgradient evaluation through Cholesky decomposition (for the Gaussian case). Indeed we only need to apply Cholesky decomposition once to a permuted sample covariance matrix to obtain each subgradient of the Lovász extension (rather than evaluating the function value O(n) times).
> - Warmstart initialization for the IP master problem. We initialize the master IP with the solution obtained from the last iteration to quickly drive up the primal bound.
>
> The updated results demonstrating these enhancements are shown in the following table and will be incorporated in the revised manuscript.
> | n, N, d       | BIC score (previous) | BIC score (updated) | CG time (previous) | CG times (updated) |
> |-----------------|--------------------------|-------------------------|--------------------------|-------------------------|
> | 15, 5000, 1   | -101955.22               | -101954.63              | 24.92s                   | 25.76s                  |
> | 15, 5000, 2   | -101516.72               | -101646.26              | 217.87s                  | 170.39s                 |
> | 15, 20000, 1  | -410315.40               | -410415.68              | 46.32s                   | 36.71s                  |
> | 15, 20000, 2  | -400946.36               | -400909.42              | 284.91s                  | 229.80s                 |
> | 25, 5000, 1  | -170335.63               | -170279.79              | 272.19s                  | 261.84s                 |
> | 25, 5000, 2  | -171048.00               | -171333.89              | 4850.47s                 | 3054.63s                |
> | 25, 20000, 1  | -683322.64               | -683475.44              | 650.32s                  | 404.43s                 |
> | 25, 20000, 2  | -688321.14               | -689041.33              | 4162.49s                 | 3189.98s                |
> |Average  |-312,540.19		   |-312,582.26		    |674.10s		     |495.62s		 |
>
> (Any n=20, d=0.5, 1.5 instance are omitted due to character limits)
>
> The reason for different BIC score: for the updated version, we employ warm-start initialization for the Master IP solution. The BIC score differences arise because the warm-start initialization causes the IP callback to find different cycles than before, which in turn selects different clusters and ultimately yields varied results.\
> Concerning the computational complexity of our row and column generation scheme for solving the Master IP, while IPs are generally not polynomial-time solvable, our framework demonstrates strong practical performance through a careful problem formulation. Although we cannot provide theoretical complexity guarantees for the pricing problem (like many practical problems solved using IP), empirical evidence suggests our method represents one of the most promising score-based approaches.
>
> **3. Tie-breaking Strategy**\
> For the tie-breaking strategy in subgradient computation, our manuscript currently implements tie-breaking up to the ranking function implemented in numpy *argsort* function. In response to the reviewer's suggestion, we've conducted additional experiments using the strategy proposed in [1]. This investigation confirms the robustness of our approach to different tie-breaking methods. In the table, 'Previous' represents our original implementation, and 'Strategic' represents that strategy in [1] is used for breaking ties.
> | n, N, d | BIC Score (Previous) | BIC Score (Strategic)|
> |----------------------|--------------------|---------------------|
> | 15, 5000, 1     | -101954.63         | -101935.35          |
> | 15, 5000, 2      | -101646.26         | -101680.72          |
> | 15, 20000, 1      | -410415.68         | -410364.40          |
> | 15, 20000, 2      | -400909.42         | -401300.69          |
> | 25, 5000, 1       | -170279.79         | -170367.37          |
> | 25, 5000, 2       | -171333.90         | -171165.26          |
> | 25, 20000, 1      | -683475.44         | -682028.55          |
> | 25, 20000, 2      | -689041.34         | -689041.34          |
> |Average      |-312,582.26		|-312,621.99		|
>
> (All n=20, d=0.5, 1.5 instances are omitted due to character limits)
>
>
> | n, N, d | Average Pricing Time (Previous) | Average Pricing Time (Strategic) |
> |----------------------|----------------|----------------|
> | 15, 5000, 1       | 0.04s          | 0.41s          |
> | 15, 5000, 2       | 0.05s          | 0.40s          |
> | 15, 20000, 1     | 0.13s          | 1.85s          |
> | 15, 20000, 2      | 0.15s          | 1.74s          |
> | 25, 5000, 1      | 0.13s          | 3.62s          |
> | 25, 5000, 2      | 0.20s          | 3.24s          |
> | 25, 20000, 1      | 0.44s          | 17.10s         |
> | 25, 20000, 2     | 0.68s          | 0.68s          |
> |Average	     |0.19s		|3.08s		|
>
> **4. Score-based IP Approach v.s. Constraint-based Approach**\
> To better motivate our score-based IP approach, we emphasize that constraint-based methods, while often more scalable in practice, operate sequentially and are susceptible to error propagation [2]. In contrast, the holistic score-based formulation avoids this issue by considering the problem globally, though at potentially greater computational cost. The IP framework transforms the original problem complexity into that of solving the optimization problem, which our method specifically aims to address. We acknowledge that constraint-based methods like PC can be more scalable for sparse data, though they may still exhibit exponential worst-case complexity [3].
>
> **5. Effect of the Supermodular Term in Scoring Function**\
> We clarify the relationship between supermodular term and DCA performance: In our current implementation, each DCA iteration optimizes a convex program formed by linearizing the concave (supermodular) component of the pricing objective. The presence of additional supermodular penalty term (for $i\gets J$) is of the form:
> $$L^i(J)=\Lambda(a_i-1)\prod_{j\in J} a_j$$
> where $\Lambda$ is a hyperparameter and $a_j,j=1\ldots,n$ is the number of values can that variable $x_j$ can take. This exponetially growing penalty term is hard to approximate using a single linear approximation.
>
> **6. Error Bars of the Results**\
> Regarding error bars, we initially omitted them due to space constraints but now include them in the table below to demonstrate the robustness of our results. We will include the complete results with error bars in the appendix of the manuscript.
> In the table below, each row represents the range for the resulting BIC score across 10 instances (generated with different modeling coefficients).
> | n, N, d      | BIC score range              |
> |--------------|-------------------------------|
> | 15, 5000, 1  | [-108960.02, -97244.45]       |
> | 15, 5000, 2  | [-105363.73, -98345.70]       |
> | 15, 20000, 1 | [-416161.99, -400230.34]      |
> | 15, 20000, 2 | [-422703.77, -389511.92]      |
> | 25, 5000, 1  | [-175506.33, -164339.46]      |
> | 25, 5000, 2  | [-176386.77, -166138.00]      |
> | 25, 20000, 1 | [-719570.68, -655633.66]      |
> | 25, 20000, 2 | [-705351.05, -664208.20]      |
>
> (GOBNILP results are omitted here due to character limits)
>
> **7. Other**\
> -We appreciate your suggestion regarding the verification of local minimality and the restart of DCA. Given the current time limitations, we will conduct further tests but are unable to include the results in the current version.\
> -The larger instances referenced in Section 5.1 were generated following the methodology in Section 5.2 with $N$=5,000. \
> -We use the Python implementation of GOBNILP for comparison in our numerical studies.\
> -Finally, we confirm that our work is primarily methodological in nature without direct social impact, and sincerely apologize for any oversight in not explicitly declaring this in the checklist.
>
> We thanks for the reviewer's insightful comments, and would be happy to provide any additional clarification or information as needed.
>
>
> ## References
> [1] Marwa El Halabi, George Orfanides, and Tim Hoheisel. Difference of sub-modular minimization via DC programming. In *International Conference on Machine Learning*, pages 9172–9201. PMLR, 2023.\
> [2] Federica Cugnata, Ron S Kenett, and Silvia Salini. Bayesian networks in survey data: Robustness and sensitivity issues. *Journal of Quality Technology*, 48(3):253–264, 2016.\
> [3] Thuc Duy Le, Tao Hoang, Jiuyong Li, Lin Liu, Huawen Liu, and Shu Hu. A fast PC algorithm for high dimensional causal discovery with multi-core PCs. *IEEE/ACM transactions on computational biology and bioinformatics*, 16(5):1483–1495, 2016.

---

> > ### Comment · Reviewer_6rfE · 2025-08-01
> >
> > Thank you for the response which addressed most of my concerns. I increased my score accordingly. I encourage the authors to include these discussions and additional results in their revision of their paper, including the faster subgradient evaluation mentioned, which is quite interesting!
> > I also encourage the authors to try the initialization strategy I suggested for DCA, it might be helpful.
> > For discrete datasets, I suggest trying the SupSub method from [30], which linearizes the submodular component in the objective then maximize the resulting submodular function. This might be better suited when the objective has a highly supermodular component.

---

> > > ### Author Response · Authors · 2025-08-01
> > >
> > > Thank you again for your insightful comments. We will make sure the discussions and additional results are included in the revision. We also believe there is still room for improvement, and will carefully test the initialization strategy and the SupSub method you suggest.

---

### Official Review · Reviewer_krir · 2025-07-02

**Clarity:** 4
**Significance:** 3
**Originality:** 3
**Rating:** 5
**Confidence:** 4

**Summary:**

The paper describes a column generation integer-programming approach for Bayesian network structure (graph) learning. The novelty is an inexact pricing method based on algorithms for difference-of-convex-function minimization; the rest of the overall framework is a recent exact method for the same problem, with exact but very costly pricing subproblems. Embedding the inexact pricing routine proposed in the paper into the column generation framework is demonstrated to provide high-quality results in typically notably shorter time than the exact approach, and the overall procedure is competitive with several other (inexact) algorithms that have been proposed for this graph learning problem over the past decade or so.

**Questions:**

- Tables 2 and 3 (and 4 and 5 in the appendix) should be complemented by the associated running times of the different methods. The current presentation does not give any information about this, but it constitutes an important aspect when discussing solution algorithms (even, maybe especially, if talking about inexact methods).
- Have the authors tested a version of their method that resorts to (costly) exact pricing if the novel heuristic pricing fails to provide a new column? That would turn the overall approach into an exact algorithm, which would still exploit the faster heuristic pricing whenever possible. It would be interesting to see how this compares to the inexact and the older exact method.
- A few points could be made a bit clearer: Could you clarify what exactly is being "observed", i.e., what the "given observational data $D$" entails? Could you state clearly how you implemented your code (is it written in R?), and did you implement the other methods yourself or use codes provided by the respective authors? (For fairness of runtime comparisons: Are they all coded in the same language, or is it a mix?) What memory and CPU speed does the employed computer have? Finally, regarding reproducibility of the experiments, will a link to the code and data be provided upon acceptance? (Otherwise, the results are not reproducible exactly, due to missing information regarding programming language, random seeds used, etc.)

* edit: I just saw the supplementary material file; somehow missed that initially. It contains the data and exact and inexact column generation approach coded in Python (and will presumably be made publicly available?), so that partly answers some of the questions in my last bullet point.

**Ethical Concerns:**

["NO or VERY MINOR ethics concerns only"]

**Final Justification:**

See comments below; overall rating retained at 5.

**Limitations:**

yes

**Paper Formatting Concerns:**

I see no formatting issues.

**Quality:**

3

**Strengths And Weaknesses:**

The paper is very well written and clearly explains the column generation approach, which is technically not the main contribution of the paper, but still commendable, especially given that most of the machine learning community is not overly familiar with the technical details of integer programming solution techniques and algorithms. The novel inexact pricing scheme is well-motived, described clearly, and then demonstrated in numerical experiments to yield an overall algorithm that is at least competitive with several others. Thus, the paper is a step forward to improved exact algorithms rooted in integer programming / column generation for the Bayesian network structure learning problem. The technical proofs also appear to be correct, as far as I could discern. The only thing I was missing is a runtime comparison -- this is only done in Table 1, but not Tables 2 and 3, i.e., only for the new method and the related exact approach, but not when comparing against other algorithms (GI, PC, MMHC).

---

> ### Author Rebuttal · Authors · 2025-07-30
>
> We sincerely appreciate your valuable feedback and insightful suggestions. Below, we address your comments in detail.
>
> **1. Runtime Comparison with Baseline Methods**\
> The runtime performance of all baseline methods is summarized in the table below. This table will be included in Appendix.
> | n, N, d      | stable PC | HC     | MMHC   |
> |---------|-----------|--------|--------|
> | 15, 5000, 0.5 | 0.038s | 0.048s | 0.048s |
> | 15, 5000, 1 | 0.035s | 0.054s | 0.047s |
> | 15, 5000, 1.5 | 0.054s | 0.101s | 0.062s |
> | 15, 5000, 2 | 0.095s | 0.098s | 0.092s |
> | 15, 20000, 0.5 | 0.097s | 0.154s | 0.126s |
> | 15, 20000, 1 | 0.121s | 0.196s | 0.152s |
> | 15, 20000, 1.5 | 0.188s | 0.263s | 0.326s |
> | 15, 20000, 2 | 0.306s | 0.420s | 0.889s |
> | 20, 5000, 0.5 | 0.050s | 0.066s | 0.090s |
> | 20, 5000, 1 | 0.105s | 0.159s | 0.112s |
> | 20, 5000, 1.5 | 0.180s | 0.309s | 0.291s |
> | 20, 5000, 2 | 0.266s | 0.342s | 0.425s |
> | 20, 20000, 0.5 | 0.146s | 0.255s | 0.185s |
> | 20, 20000, 1 | 0.279s | 0.481s | 0.438s |
> | 20, 20000, 1.5 | 0.660s | 0.950s | 1.366s |
> | 20, 20000, 2 | 1.327s | 2.331s | 3.634s |
> | 25, 5000, 0.5 | 0.083s | 0.099s | 0.071s |
> | 25, 5000, 1 | 0.095s | 0.133s | 0.098s |
> | 25, 5000, 1.5 | 0.227s | 0.348s | 0.218s |
> | 25, 5000, 2 | 0.273s | 0.403s | 0.351s |
> | 25, 20000, 0.5 | 0.200s | 0.407s | 0.235s |
> | 25, 20000, 1 | 0.288s | 0.637s | 0.583s |
> | 25, 20000, 1.5 | 0.725s | 1.568s | 2.612s |
> | 25, 20000, 2 | 1.158s | 2.087s | 2.940s |
>
> We acknowledge that certain constraint-based methods exhibit significantly faster execution times. Compared to constraint-based method that operate sequentially and are susceptible to error propagation [1], our score-based approach considers the problem globally, though at potentially greater computational cost. Compared with the Hill Climbing (HC) method that aims at heuristically solving the problem at a low cost, the score-based IP methods transform the original problem complexity into that of solving the optimization problem, which our method specifically aim to addresses. Furthermore, our ultimate goal is to develop a truly exact score-based method, where our current framework serves as a subroutine to efficiently generate high-quality columns.
>
> **2. Exact Pricing Method**\
> We greatly appreciate your suggestion regarding the use of exact pricing. In response, we are currently developing an enhanced solution approach by incorporating the cutting plane method proposed in [2]. Unlike solving the pricing MINLP directly with Gurobi—which introduces numerical errors due to logarithmic term approximations—our exact solution avoids this issue.  This exact pricing method is activated when the heuristic algorithm fails to identify new columns. Below, we present some preliminary results obtained by integrating exact pricing into our framework (still using the restricted master heuristic though):
> | n, N, d   | Average BIC score (exact) | Average BIC score (heuristic) | Average Time (exact) | Average Time (heuristic) |
> |---------------------------------|------------------------|----------------------------|---------------------------|-------------------------------|
> | 15, 5000, 0.5                |       -101818.64             | -101818.64                 | 1738.09s                   | 4.06s                          |
> | 15, 5000, 1                  |           -101939.28             | -101954.63                 | 6923.13s                   | 25.76s                         |
> | 15, 5000, 1.5                  |            -101350.15             | -101411.14                 | 11597.61s                  | 76.72s                         |
> | 15, 5000, 2                   |         -101414.39             | -101646.26                 | 11903.48s                  | 170.39s                        |
>
> While these initial results demonstrate the effectiveness of the exact method in improving the objective value, we observe that the solution time scales with problem complexity, which motivates ongoing research into more efficient implementations. Our current efforts focus on optimizing the exact pricing framework to maintain its advantages while improving scalability for larger instances.
>
> **3. Clarification on Observational Data**\
> We apologize for any confusion regarding the observational data notation. To clarify, the dataset $D$
> is  $N \times n$
>  data matrix where:\
>     - $N$ : number of data samples\
>     - $n$ : number of random variables (nodes in the Bayesian network)\
>     - Each row $x_i = (x_{i,1},...,x_{i,n})$ represents a complete data sample\
>     - Each column $D_{:,j} = (x_{1,j},...,x_{N,j})^\top$ represents all observations for the $j$-th variable\
>     - $x_{i,j}$ = value of the $j$-th variable in the $i$-th observation. For discrete labeled data, each label is mapped to a distinct numerical value.\
>     We hope this clarification resolves any ambiguity, and we will clarify it in the paper.
>
>  Thank you again for your constructive feedback, which helps us improve our work. We remain open to any further questions or suggestions.
>
>
>
>
> ## References
> [1] Federica Cugnata, Ron S Kenett, and Silvia Salini. Bayesian networks in survey data: Robustness and sensitivity issues. *Journal of Quality Technology*, 48(3):253–264, 2016.\
> [2] George L Nemhauser and Laurence A Wolsey. Maximizing submodular set functions: formulations and analysis of algorithms. In *North-Holland Mathematics Studies*, volume 59, pages 279–301. Elsevier, 1981.

---

> > ### Comment · Reviewer_krir · 2025-08-04
> >
> > I thank the authors for the detailed responses to the questions raised by the other reviewers and myself. I feel that all main concerns have been addressed adequately and trust that the authors will also take care to clarify the few small points that were not explicitly referenced in their rebuttals (such as computer architecture specifics, or a clarification regarding which methods were implemented in which language, R or Python; this can be gleaned from reading all rebuttals, but of course would be good to state once and clearly in the paper). I already gave the paper a clear "accept" score, which I will certainly retain.

---

> > > ### Author Response · Authors · 2025-08-04
> > >
> > > Thank you again for your kind comments. We will make sure those missing details will be clarified in the paper as well.

---

### Official Review · Reviewer_oPou · 2025-07-05

**Clarity:** 3
**Significance:** 2
**Originality:** 3
**Rating:** 5
**Confidence:** 2

**Summary:**

This paper addresses the computational bottleneck in Bayesian Network Structure Learning (BNSL) via Integer Programming (IP), where traditional column generation (CG) suffers from expensive pricing problems solved via Mixed-Integer Nonlinear Programming (MINLP). The authors propose CG-DCA: a CG framework that reformulates pricing as a Difference-of-Submodular (DS) optimization problem, solved approximately via the Difference-of-Convex Algorithm (DCA).

**Questions:**

1. Could you propose/explore alternative DS decompositions or scoring functions to handle the super-modular penalty in discrete BNSL? Would DCA with tailored sub-gradients help?

2. How sensitive is DCA to initialization? Does the hybrid strategy consistently outperform pure warm-start/random?

**Ethical Concerns:**

["NO or VERY MINOR ethics concerns only"]

**Final Justification:**

All my concerns have been addressed.

**Limitations:**

yes

**Quality:**

2

**Strengths And Weaknesses:**

Strengths:
1. Novelty: First to reformulate BNSL pricing as DS optimization, creatively leveraging DCA’s efficiency. Theoretically grounded (Propositions 1–2).
2. Practical Impact: Achieves 10–100× speedup vs. MINLP pricing (0.3s vs. 35.1s per node) and higher solution quality than GOBNILP on Gaussian data (Table 1). Handles denser graphs d=2 better than exact IP.
3. Clarity:  Strong exposition of BNSL IP formulation (Section 2.1), CG pitfalls, and DS optimization.

Weaknesses:
1. Limited Scope:  Effectiveness is restricted to Gaussian data. Discrete-data performance degrades severely (Table 3), acknowledged but not mitigated.
2. Scalability Gaps: Runtime remains high for n=25,d=2 (4,850s), and no results for n>37. DCA’s convergence in pricing is empirical but untheorized.
3. Baseline Comparison:** Lacks runtime comparison against non-IP baselines (PC, MMHC). Some metrics (SHD) show PC/MMHC competitive or better (Table 2).

---

> ### Author Rebuttal · Authors · 2025-07-30
>
> We thank the reviewer for their time and insightful feedback. Our responses to the comments are provided below.
>
> **1. Limitations in Discrete Data Implementation**\
> We acknowledge the limitations of our current implementation for discrete data. Regarding the decomposition of the discrete BIC score, the approach we employ represents the most effective strategy we have identified so far, though we recognize that alternative decompositions may exist.\
> To address the supermodular penalty issue in discrete BNSL, we are currently investigating a strategy that involves convexifying the cardinality penalty term in the BIC score. To be specific, the pricing problem objective for node $i$ can be decomposed as $F^i(J) = G^i(J) - H^i(J) + L^i(J)$, where both $G^i(J)$ and $H^i(J)$ are submodular, and the supermodular penalty term $L^i(J)=\Lambda(a_i-1) \prod_{j\in J} a_j$ depends on the number of possible values $a_j$ for each discrete variable $X_j$ , $j=1,\ldots,n$. In our difference-of-submodular formulation, $G^i(J)$ represents the first submodular function, while $H^i(J) - L^i(J)$ constitutes the second. Unlike previous implementations that iteratively linearize the whole supermodular component $(-H^i(J)+ L^i(J))$ in DCA, leading to higher approximation errors, the new approach currently under investigation convexifies the penalty term $L^i(J)$ through cutting planes while maintaining $G^i(J) - H^i(J)$ in the DC program. Assuming binary data (such that $L^i(J)=\Lambda 2^{|J|}$), we enhance the optimization program by adding cutting planes that represent the convex envelope of the supermodular penalty term $L^i(J)$. We have a preliminary experiment result on LUCAS data: \
> Precision: 0.85, Recall: 0.92, SHD: 2,\
> and may improve and implement the approach to more general settings.\
> Additionally, we have developed a decomposition strategy for the BDeu score as a difference of submodular set functions. This approach may have potential to help mitigate the supermodularity-related difficulties, due to the structural differences between the BDeu and discrete BIC scores. While implementation is still underway, we plan to explore this in future work and will leave some relevant ideas in the conclusion section of this manuscript.\
> Regarding the tailored subgradient method, we are not sure what you are referring to and have not yet incorporated it into our framework. However, we would greatly appreciate any references or additional information you could provide to guide its potential implementation.
>
> **2. Runtime Comparison with Baseline Methods**\
> The runtime performance of all baseline methods (implemented in R) is summarized in the table below. This table will be included in Appendix.
> | n, N, d      | stable PC | HC     | MMHC   |
> |---------|-----------|--------|--------|
> | 15, 5000, 0.5 | 0.038s | 0.048s | 0.048s |
> | 15, 5000, 1 | 0.035s | 0.054s | 0.047s |
> | 15, 5000, 1.5 | 0.054s | 0.101s | 0.062s |
> | 15, 5000, 2 | 0.095s | 0.098s | 0.092s |
> | 15, 20000, 0.5 | 0.097s | 0.154s | 0.126s |
> | 15, 20000, 1 | 0.121s | 0.196s | 0.152s |
> | 15, 20000, 1.5 | 0.188s | 0.263s | 0.326s |
> | 15, 20000, 2 | 0.306s | 0.420s | 0.889s |
> | 20, 5000, 0.5 | 0.050s | 0.066s | 0.090s |
> | 20, 5000, 1 | 0.105s | 0.159s | 0.112s |
> | 20, 5000, 1.5 | 0.180s | 0.309s | 0.291s |
> | 20, 5000, 2 | 0.266s | 0.342s | 0.425s |
> | 20, 20000, 0.5 | 0.146s | 0.255s | 0.185s |
> | 20, 20000, 1 | 0.279s | 0.481s | 0.438s |
> | 20, 20000, 1.5 | 0.660s | 0.950s | 1.366s |
> | 20, 20000, 2 | 1.327s | 2.331s | 3.634s |
> | 25, 5000, 0.5 | 0.083s | 0.099s | 0.071s |
> | 25, 5000, 1 | 0.095s | 0.133s | 0.098s |
> | 25, 5000, 1.5 | 0.227s | 0.348s | 0.218s |
> | 25, 5000, 2 | 0.273s | 0.403s | 0.351s |
> | 25, 20000, 0.5 | 0.200s | 0.407s | 0.235s |
> | 25, 20000, 1 | 0.288s | 0.637s | 0.583s |
> | 25, 20000, 1.5 | 0.725s | 1.568s | 2.612s |
> | 25, 20000, 2 | 1.158s | 2.087s | 2.940s |
>
> We acknowledge that certain constraint-based methods exhibit significantly faster execution times. Compared to constraint-based method that operate sequentially and are susceptible to error propagation [1], exact score-based approaches consider the problem globally, though at potentially greater computational cost. Compared with the Hill Climbing (HC) method that aims at heuristically solving the problem at a low cost, the score-based IP methods transform the original problem complexity into that of solving the optimization problem, which our method specifically aim to addresses. Furthermore, our ultimate goal is to develop a truly exact score-based method, where our current framework serves as a subroutine to efficiently generate high-quality columns. Please refer to our response to Reviewer krir for some additional discussions.
>
> **3. Sensitivity to DCA Initialization**\
> To evaluate the sensitivity of DCA initialization, we compare the BIC scores and time costs across three initialization methods (warm-start, random, and hybrid). The results are summarized in the following tables:
> | n, N, d      | hybrid       | random       | warmstart    |
> |--------------|--------------|--------------|--------------|
> | 15, 5000, 0.5 | -101818.64   | -101818.64   | -101819.58   |
> | 15, 5000, 1   | -101954.63   | -101939.75   | -101975.70   |
> | 15, 5000, 1.5 | -101411.14   | -101376.34   | -101509.59   |
> | 15, 5000, 2   | -101646.26   | -101556.54   | -101608.25   |
> | 15, 20000, 0.5 | -406345.25   | -406345.25   | -406346.23   |
> | 15, 20000, 1  | -410415.68   | -410415.68   | -410382.11   |
> | 15, 20000, 1.5 | -405344.24   | -405487.19   | -405599.33   |
> | 15, 20000, 2  | -400909.42   | -400861.51   | -401710.97   |
> | 20, 5000, 0.5 | -136605.28   | -136605.28   | -136734.74   |
> | 20, 5000, 1   | -137222.13   | -137212.97   | -137363.27   |
> | 20, 5000, 1.5 | -136624.81   | -136577.03   | -136957.87   |
> | 20, 5000, 2   | -136849.69   | -136610.04   | -136750.44   |
> | 20, 20000, 0.5 | -535047.92   | -535047.92   | -535183.20   |
> | 20, 20000, 1  | -541850.16   | -541887.02   | -542168.27   |
> | 20, 20000, 1.5 | -538682.19   | -538693.99   | -539863.90   |
> | 20, 20000, 2  | -535301.38   | -534115.87   | -534939.03   |
> | 25, 5000, 0.5 | -170751.53   | -170751.53   | -170751.74   |
> | 25, 5000, 1   | -170279.79   | -170296.43   | -170302.01   |
> | 25, 5000, 1.5 | -171308.77   | -171157.64   | -171289.38   |
> | 25, 5000, 2   | -171333.89   | -171105.48   | -170760.08  |
> | 25, 20000, 0.5 | -677813.87   | -677813.87   | -677812.65   |
> | 25, 20000, 1  | -683475.44   | -683189.30   | -683826.26   |
> | 25, 20000, 1.5 | -676190.15   | -676267.45   | -676649.76   |
> | 25, 20000, 2  | -689041.33   | -688690.44   | -687147.17  |
> | **Average**   | **-312,582.26** | **-312,553.87** | **-312,712.62** |
>
> | n, N, d      | hybrid    | random    | warmstart |
> |--------------|-----------|-----------|-----------|
> | 15, 5000, 0.5 | 4.06s     | 4.04s     | 1.17s     |
> | 15, 5000, 1   | 25.76s    | 29.40s    | 4.49s     |
> | 15, 5000, 1.5 | 76.72s    | 87.43s    | 10.00s    |
> | 15, 5000, 2   | 170.39s   | 295.25s   | 23.74s    |
> | 15, 20000, 0.5 | 10.17s    | 10.94s    | 3.51s     |
> | 15, 20000, 1  | 36.71s    | 36.57s    | 10.63s    |
> | 15, 20000, 1.5 | 104.25s   | 109.73s   | 23.95s    |
> | 15, 20000, 2  | 229.80s   | 245.11s   | 41.94s    |
> | 20, 5000, 0.5 | 9.53s     | 9.60s     | 2.42s     |
> | 20, 5000, 1   | 84.34s    | 90.87s    | 12.51s    |
> | 20, 5000, 1.5 | 312.67s   | 430.37s   | 48.78s    |
> | 20, 5000, 2   | 1039.53s  | 1770.71s  | 95.36s    |
> | 20, 20000, 0.5 | 18.42s    | 18.54s    | 8.71s     |
> | 20, 20000, 1  | 133.69s   | 134.79s   | 34.84s    |
> | 20, 20000, 1.5 | 425.17s   | 504.29s   | 444.37s   |
> | 20, 20000, 2  | 997.99s   | 1710.91s  | 211.44s   |
> | 25, 5000, 0.5 | 21.97s    | 22.35s    | 3.87s     |
> | 25, 5000, 1   | 261.84s   | 304.12s   | 33.59s    |
> | 25, 5000, 1.5 | 949.90s   | 2434.03s  | 131.69s   |
> | 25, 5000, 2   | 3054.63s  | 6037.58s  | 224.73s  |
> | 25, 20000, 0.5 | 50.22s    | 53.13s    | 12.48s    |
> | 25, 20000, 1  | 404.43s   | 642.87s   | 78.21s    |
> | 25, 20000, 1.5 | 1372.59s  | 3688.33s  | 227.15s   |
> | 25, 20000, 2  | 3189.98s  | 6674.62s  | 366.46s  |
> | **Average**   | **495.62s** | **897.93s** | **88.71s** |
>
> From the table, we can see that:
>
> - Random initialization yields highest (best among three)
> BIC scores in 11 out of 24 instances among three methods but requires the most time in 22 out of 24 instances.
>
> - Warm-start initialization achieves the lowest (worst among three) BIC scores in 16 out of 24 instances and is the most time-efficient in 23 out of 24 instances.
>
> - The hybrid approach strikes a balance between optimality and computational efficiency, since it focuses on local refinement around the current best pattern (through warm-start initialization) while building upon the foundation of global exploration (through random initialization) at the early stage.
>
> We appreciate the opportunity to clarify these points and remain open to further discussion or suggestions.
>
> ## References
> [1] Federica Cugnata, Ron S Kenett, and Silvia Salini. Bayesian networks in survey data: Robustness and sensitivity issues. *Journal of Quality Technology*, 48(3):253–264, 2016.

---

### Decision · Program_Chairs · 2025-09-17

**Decision:**

Accept (poster)

**Comment:**

This paper introduces an inexact column generation framework for Bayesian Network Structure Learning (BNSL). In particular, the pricing problem is reformulated as a difference-of-submodular optimization problem and solved approximately with the Difference-of-Convex Algorithm (DCA). Empirical results demonstrate that the proposed method provides significant speedups over exact column generation and yields higher-quality solutions on Gaussian data compared with state-of-the-art score-based approaches.

Overall, this paper reads as an excellent fit to NeurIPS: it is novel within the BNSL literature and well motivated. The presentation is also rigorous and makes the paper particularly relevant both to ML and optimization audiences. The review team also appreciated the theoretical grounding of the DS reformulation and the clarity of exposition.

There were, however, important points raised during the discussion. Specifically, performance degrades substantially on discrete data, scalability is still restricted to relatively small graph sizes, and the heuristic nature of DCA and its structure make it difficult to provide strong theoretical guarantees. Initial reviewer concerns centered on runtime comparisons, initialization strategies, and experimental clarity.

Nonetheless, the author-reviewer discussions were very productive, and all questions were addressed nicely in the rebuttal with additional results and clarifications. I must note that all are minor and can be easily incorporated into the current draft. I appreciate the author-reviewer engagement and the thorough authors' responses.

Given the combination of novelty, technical soundness, and demonstrated benefits, the review team supports the acceptance of the paper at NeurIPS. We all believe it is an impactful work, specifically by providing a valuable contribution at the interface of optimization and causal discovery.